# GENERALIZED RESOURCE-AWARE DISTRIBUTED MINIMAX OPTIMIZATION

## ABSTRACT

Traditional distributed minimax optimization algorithms cannot be applied in resource-limited clients dealing with large-scale models. In this work, we present *SubDisMO*, a generalized resource-aware distributed minimax optimization algorithm. *SubDisMO* prunes the global large-scale model into adaptive-sized submodels to accommodate varying resources during each communication round. However, the randomly pruned submodels are susceptible to *arbitrary submodel sharpness*, which can hinder generalization and lead to slow convergence. To address this issue, *SubDisMO* trains the arbitrarily pruned submodels with perturbations by optimizing the minimax objectives, enhancing the *generalization* performance of the aggregated full model. We theoretically analyze our proposed resource-aware *SubDisMO* algorithm, demonstrating that it achieves an asymptotically optimal convergence rate of $O(1/\sqrt{QTC^*})$, which is dominated by the minimum covering number $C^*$. We also show the generalization bound of *SubDisMO* corresponding to the perturbation and parameter remaining rate in each layer. Extensive experiments on *CIFAR-10* and *CIFAR-100* datasets demonstrate that *SubDisMO* achieves superior generalization and effectiveness compared to state-of-the-art baselines.

## 1 INTRODUCTION

Recently, distributed minimax problem has gained tremendous popularity due to the concerns on privacy and security and the model optimization on edge. SGDAM-PEF (Zhang et al., 2023), LocalSCGDAM (Zhang et al., 2024) formulate the Area-Under-the-ROC-Curve (AUC) maximization problem as a federated compositional minimax optimization problem. Distributionally robust optimization (DRO) problem (Sinha et al., 2018; Deng et al., 2020; Zhu et al., 2024) aims to find solutions that perform well under the worst-case scenario within a predefined set of possible probability distributions, enhancing robustness against distributional uncertainty. FedSAM (Qu et al., 2022), FedGDA-GT (Sun & Wei, 2022) and FedSGDA+ (Wu et al., 2023) focus on provably optimizing the following distributed minimax problem,

$$\min_{\theta} \max_{\delta} \left\{ f(\theta, \delta) = \frac{1}{N} \sum_{i \in [N]} f_i(\theta, \delta) \right\}, \tag{1}$$

where $N$ denotes the number of clients, $f_i(\theta, \delta) = \mathbb{E}_{\xi_i \sim D_i}[f_i(\theta, \delta; \xi_i)]$ is the local loss function, and $\theta \in \mathbb{R}^{d_\theta}$ and $\delta \in \mathbb{R}^{d_\delta}$.

However, with the arising of large-scale models (Jiao et al., 2023; Zhou et al., 2023a; Min et al., 2023), the size of full model $\theta$ is tremendous and it is hard to run in the resource-limited clients. Thus, traditional aggregation mechanisms cannot be applied directly. Therefore, a great deal of work has been proposed, such as RAM-Fed (Wang et al., 2023), OAP (Zhou et al., 2023b), IST (Yuan et al., 2022), PruneFL (Jiang et al., 2022), to reduce the scale of the large-scale model and communication cost, so that they can be adapted to varying resources. However, the mentioned methods mainly aim at *minimization optimization* by adopting a gradient descent to find the local minima traditionally, failing to solve the mentioned *minimax optimization*. Thus in this work, we consider the resource-aware

distributed minimax optimization problem as follows:

$$\min_{\theta} \max_{\|\epsilon_i\| \leq \delta} \left\{ f(\theta, \epsilon) = \frac{1}{N} \sum_{i \in [N]} f_i(\theta, \epsilon_i) \right\},$$

$$\text{s.t.} \quad f_i(\theta, \epsilon_i) = \max_{\|\epsilon_i\| \leq \delta} f_i(\theta \odot m_i, \epsilon_i), \quad \forall i \in [N],$$

(2)

where $\delta$ is a predefined constant controlling the perturbation. During local training, clients actually train the submodel $\theta \odot m_i$, where $m_i$ is the local mask. Specifically, $m_i$ can be changed over time so that the client can train the submodel adapted to local dynamic resources.

It's challenging to solve the above mentioned distributed minimax optimization problem in two aspects. *1) On the algorithm side,* when training the random submodel in clients, i.e. in Figure 1, client C trains $\theta_b$, the original minimize optimization problem may fall into *arbitrary submodel sharpness* due to overfitting to the local distribution. Thus, when aggregated overlapped partial parameters, i.e. in Figure 1, $\theta_a$ in client A and B, global model could be inconsistent and divergent, which will degrade the performance of the global model and even slow down the model convergence speed. Naturally, we consider that if we reduce the level of *arbitrary submodel sharpness* at local minima, even

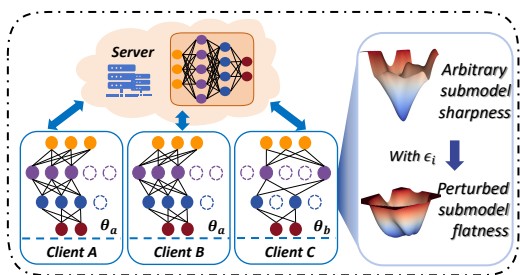

Figure 1: The submodel training paradigm and the comparison between the origin arbitrary submodel sharpness and the perturbed submodel flatness loss landscape.

adding a bit perturbation, i.e. $\theta_b + \epsilon$, to make the aggregated model generalized. *2) On the theoretical analysis side,* the interaction between minimization and maximization subproblems complicates the theoretical analysis in both the convergence speed and the generalization performance. It's essential to explore the key factors affecting the result of both the convergence speed and the generalization performance to guide the algorithm design in resource-aware distributed minimax optimization.

To solve this kind of distributed resource-aware minimax optimization problem while improving the generality of the global model, we design a new distributed learning algorithm that adaptively generates submodels through local resources and trains with perturbations namely *SubDisMO*. We introduce an additional gradient ascent process to approximate linear constrained inner maximization, then use local stochastic gradient descent. Thus, we can minimize the worst loss of the perturbed submodel in local training. After receiving all submodels' updates, the server aggregates them to update the global model. *Theoretically, 1) we establish an asymptotically optimal convergence rate $\mathcal{O}(1/\sqrt{QTC^*})$ of our algorithm*, where $Q$ is the communication rounds, $T$ the local iteration and $\mathcal{C}^*$ is the minimum covering number defined in Section 4. *2) From the generalization aspect, we give a tighter error bound (shown in Theorem 2)* corresponding to the perturbation $\delta$ and parameter remaining rate $s_j$ in $j$-th layer. The extensive experimental results confirm the average performance and generality of *SubDisMO*. Code is available at https://anonymous.4open.science/r/SubDisMO/.

Our contributions can be summarized as follows:

- To the best of our knowledge, we are the first to design a resource-aware distributed minimax optimization algorithm, namely *SubDisMO*. Specifically, *SubDisMO* trains the resource-adaptive submodels with perturbations to mitigate the arbitrary submodel sharpness, thereby enhancing the generalization of the global full model.

- We theoretically analyze the convergence rate and the generalization bound of *SubDisMO*. We prove that it can achieve an asymptotically optimal convergence rate $\mathcal{O}(1/\sqrt{QTC^*})$ under the non-convex condition. We give a tighter generalization bound corresponding to the perturbation and parameter remaining rate in each layer.

- We conduct extensive experiments on *CIFAR-10* and *CIFAR-100* by comparing with state-of-the-art resource-limited training paradigm. Results demonstrate the generalization and effectiveness of *SubDisMO* is better than other state-of-the-art baselines.

In summary, *SubDisMO* gives a new insight into solving distributed minimax optimization. We rigorously provide a theoretical convergence guarantee. Existing studies would be special cases of our *SubDisMO*. When $\delta = 0$ that is the perturbation is zero, the minimax optimization degrades to minimization, and the convergence rate of *SubDisMO* is identical to *RAM-Fed* (Wang et al., 2023). When $\mathcal{C}^* = N$ that is all the clients train the full model, *SubDisMO* achieves the same convergence rate $\mathcal{O}(1/\sqrt{QTN})$ as *FedSAM* (Qu et al., 2022). When $\mathcal{C}^* = N$ and $\delta = 0$ that is all the clients train the full model without perturbation, the learning paradigm degrades to *FedAvg* (McMahan et al., 2017) and achieves the same convergence rate $\mathcal{O}(1/\sqrt{QTN})$. When $\mathcal{C}^* = 1$ and $\delta = 0$ that is each client trains definitely non-overlapping submodel without perturbation, *SubDisMO* achieves the same convergence rate $\mathcal{O}(1/\sqrt{QT})$ as *OAP* (Zhou et al., 2023b). Otherwise, we are the first to give a generalization error bound in resource-limited scenarios and we establish the impact of perturbation $\delta$ and parameter remaining rate $s_j$ on it. When $s_j = 1$ that is each client trains the full model, the generalization bound is identical to *FedSAM* (Qu et al., 2022)

## 2 RELATED WORK

Distributed minimax optimization has seen significant advancements driven by the need to handle large-scale and complex problems efficiently. In order to solve the impact of imbalanced data, Ying et al. (2016) directly optimizes the Area-Under-the-ROC-Curve (AUC) score instead of cross-entropy loss function and formulates it as a minimax optimization problem. SGDAM-PEF and SGDAM-REF (Zhang et al., 2023) use stochastic gradient descent ascent algorithms and consider reducing the communication cost at the same time. In addition, LocalSCGDAM (Zhang et al., 2024) develops a local stochastic compositional gradient descent ascent with momentum algorithm. Otherwise, the distributionally robust optimization (DRO) problem (Sinha et al., 2018; Deng et al., 2020) which aims to minimize the worst case in the predefined possible probability distributions has gained great attention. Recently, in order to address general federated minimax problems, Deng & Mahdavi (2021) introduce local Stochastic Gradient Descent Ascent (SGDA), which enables each device to perform multiple GDA steps before communication. The authors demonstrated sub-linear convergence for local SGDA with diminishing step sizes. Based on this, FedGDA-GT (Sun & Wei, 2022) further proposes federated gradient descent ascent with gradient tracking and proves that FedGDA-GT converges linearly with a constant stepsize to global $\epsilon$-approximation solution with $\mathcal{O}(\log(1/\epsilon))$ rounds of communication, which matches the time complexity of centralized GDA method. Wu et al. (2023) design stochastic gradient decent ascent methods FedSGDA+ and FedSGDA-M with better sample and communication complexities to match the convergence rate of single-machine. However, the existing distributed minimax optimization algorithms require sufficient computing and communication resources on clients to train the full model, without considering resource-limited scenarios, which is the main goal of our work.

## 3 METHODOLOGY

In order to alleviate the impact of *arbitrary submodel sharpness* and improve the generalization of the global full model, we propose a resource-aware distributed minimax optimization algorithm named *SubDisMO*, which can train adaptive-sized submodels in different kinds of clients and gain a generalized global model. The whole process is shown in Algorithm.1 and we give a further description in this section.

From the start of the process, the server sends the latest global model $\theta_q$ to clients, and each client use the resource-aware adaptive mask policy $P(\theta_q; R_n)$ to generate a local mask, where $R_i$ represents the resource constraints of client $i$. The mask $m_{q,n} \in \{0,1\}^{|\theta_q|}$, where each element is a binary value that determines whether a corresponding parameter in the global model $\theta_q$ is included in the client's submodel $\theta_{q,i}$. Thus the submodel trained locally by client $i$ can be expressed as,

$$\theta_{q,n,0} = \theta_q \odot m_{q,n}, \quad m_{q,n} = P(\theta_q; R_n), \tag{3}$$

where $\odot$ means the element-wise multiplication, only non-zero parameters continue to be trained, We define the set of whole parameters as $\mathcal{S}$, trained parameters as $\mathcal{K}_q$, and untrained parameters as $\mathcal{S} - \mathcal{K}_q$. This mask can be changed in different communication rounds $q$ for any client $n$ which introduces the submodel heterogeneous in our algorithm.

Then each client trains the submodel using local data. For the purpose of generalization, we consider adding the perturbation to the local submodel. Return to the objective that we want to optimize,

$$\min_\theta \max_{\|\epsilon_i\| \le \delta} \{f(\theta, \epsilon) = \frac{1}{N} \sum_{i \in [N]} f_i(\theta, \epsilon_i)\}, \quad (4)$$

where $f_i(\theta, \epsilon) \triangleq \max_{\|\epsilon_i\| \le \delta} f(\theta + \epsilon_i)$, we use the first order Taylor expansion to approximate it and gain the perturbed model $\tilde{\theta}$ for epoch $t = 1$ to $T$:

$$\tilde{\theta}_{q,n,t-1} = \theta_{q,n,t-1} + \delta \frac{g_{q,n,t-1}}{\|g_{q,n,t-1}\|}, \quad (5)$$

where $g_{q,n,t-1} = \nabla f_n(\theta_{q,n,t-1}, \xi_{n,t-1}) \odot m_{q,n}$, $\xi_{n,t-1}$ is a data sample. Here we mask the gradient as well to prevent extra value on untrained parameters. After getting the perturbed model which has the highest loss within neighborhood, we implemented the normal gradient descent algorithm to complete the model update,

$$\theta_{q,n,t} = \theta_{q,n,t-1} - \eta_l \tilde{g}_{q,n,t-1}, \quad (6)$$

where $\eta_l$ is the local learning rate and $\tilde{g}_{q,n,t-1} = \nabla f_n(\tilde{\theta}_{q,n,t-1}, \xi_{n,t-1}) \odot m_{q,n}$, $\xi_{n,t-1}$ is a data sample. So that we complete the local submodel update based on the perturbation point $\tilde{\theta}_{q,n,t-1} + \epsilon_{q,n,t-1}$.

---

**Algorithm 1:** *SubDisMO*

**Initialize:** Dataset $\mathcal{D}_n$ on $N$ clients, mask policy $P(\cdot)$, global model $\theta_1$, perturbation upper bound $\delta$

**for** *round $q = 1$ to $Q$* **do**
  **for** *$n = 1$ to $N$ (all workers in parallel)* **do**
    Generate mask $m_{q,n} = P(\theta_q, n)$
    Generate submodel $\theta_{q,n,0} = \theta_q \odot m_{q,n}$
    *# Update local submodel with perturbation:*
    **for** *epoch $t = 1$ to $T$* **do**
      Compute a local training estimate
      $g_{q,n,t-1} = \nabla f_n(\theta_{q,n,t-1}, \xi_{n,t-1}) \odot m_{q,n}$
      $\tilde{\theta}_{q,n,t-1} = \theta_{q,n,t-1} + \delta \frac{g_{q,n,t-1}}{\|g_{q,n,t-1}\|}$
      Compute a local training estimate
      $\tilde{g}_{q,n,t-1} = \nabla f_n(\tilde{\theta}_{q,n,t-1}, \xi_{n,t-1}) \odot m_{q,n}$
      $\theta_{q,n,t} = \theta_{q,n,t-1} - \eta_l \tilde{g}_{q,n,t-1}$
    **end**
    $\Delta_{q,n} = \theta_{q,n,0} - \theta_{q,n,T}$
  **end**
  *# Update all parameters of global model:*
  **for** *parameters $i = 1$ to $\mathcal{S}$* **do**
    Find $N_q^i = \{n : m_{q,n}^i = \mathbf{1}\}$
    **if** $\mathcal{C}_q^i = 0$ **then**
      Update $\theta_{q+1}^i = \theta_q^i$
    **else**
      $\Delta_q^i = \frac{1}{\mathcal{C}_q^i} \sum_{n \in N_q^i} \Delta_{q,n}^i$
      Update $\theta_{q+1}^i = \theta_q^i - \eta_g \Delta_q^i$
    **end**
  **end**
**end**

---

After the local training, each client calculate the final local updates $\Delta_{q,n} = \theta_{q,n,0} - \theta_{q,n,T}$ and upload it to the server. After the server collects all clients updates, it aggregates them by parameter. For every parameter $i$, the server calculate the number of clients that trained $i$ denoted as $\mathcal{C}_q^i$. If $\mathcal{C}_q^i = 0$, the parameter $i$ has not been trained, the parameter $\theta_q^i$ is remained. Otherwise, the server calculate the aggregate updates and update the parameter $i$ with it:

$$\theta_{q+1}^i = \theta_q^i - \eta_g \frac{1}{\mathcal{C}_q^i} \sum_{n \in N_q^i} \Delta_{q,n}^i, \quad (7)$$

where $\eta_g$ is the learning rate in server.

## 4 THEORETICAL ANALYSIS

In this section, we analyze both the convergence rate and generalization bound of the proposed *SubDisMO* and explore the impact of different key factors. First, we adopt the following commonly used in distributed learning convergence analysis:

**Assumption 1** (L-smooth). *Every function $f_n(\cdot)$ is L-smooth for all $n \in [N], \theta, \phi \in \mathbb{R}^d$,*
$$\|\nabla f_n(\theta) - \nabla f_n(\phi)\| \le L\|\theta - \phi\|. \quad (8)$$

**Assumption 2** (Bounded data heterogeneity level). *The effect of data heterogeneity level can be bounded by $\sigma_g^2$ for all $n \in [N]$,*
$$\|\nabla f_n(\theta_q) - \nabla f(\theta_q)\|^2 \le \sigma_g^2. \quad (9)$$

**Assumption 3** (Bounded variance of stochastic gradient). *The stochastic gradient $\nabla f_n(\theta_{q,n,t}, \xi_{n,t})$, computed by using mini-batch $\xi_{n,t}$ is an unbiased estimator $\nabla F_n(\theta_{q,n,t})$ bounded by $\sigma_l^2$,*
$$\mathbb{E}_{\xi_n \sim D_n} \left\| \frac{\nabla f_n(\theta_{q,n,t}, \xi_{n,t})}{\|\nabla f_n(\theta_{q,n,t}, \xi_{n,t})\|} - \frac{\nabla f_n(\theta_{q,n,t})}{\|\nabla f_n(\theta_{q,n,t})\|} \right\|^2 \le \sigma_l^2, \quad (10)$$

$\forall n \in [N]$, *where the expectation is over all local datasets.*

And this assumption is tighter than the similar assumption to bound the stochastic gradient variance, that is $\mathbb{E}_{\xi_{n,t} \sim D_n} \|\nabla f_n(\theta_{q,n,t}; \xi_{n,t}) - \nabla f_n(\theta_{q,n,t})\|^2 \leq \sigma^2$. It's obvious that $\sigma_l^2$ should be less than $\pi^2$, the norm of difference in unit vectors that can be bounded by the arc length on a unit circle.

**Assumption 4** *(Bounded noise induced from mask). The deviation of the masked parameters in client $n$ from the original parameters for every round $q$ is limited by $l \in [0,1)$:*

$$\|\theta_q - \theta_q \odot m_{q,n}\|^2 \leq l^2 \|\theta_q\|^2. \tag{11}$$

### 4.1 Convergence analysis of SubDisMO

**Definition 1** *(Minimum covering number). The minimum number of submodels training the corresponding $i$-th parameter in all rounds is defined as:*

$$\mathcal{C}^* = \min_{q,i} \mathcal{C}_{q,i}, i \in \mathcal{K}_q, \forall q, \tag{12}$$

*where $\mathcal{C}_{q,i}$ is the number of the client that train the $i$-th parameter in the communication round $q$. For full model training federated learning framework, i.e., FedAvg, $\mathcal{C}^* = N$, that is all clients train every parameter in every communication round.*

**Lemma 1** *(Bounded $\mathcal{E}_g$ (Qu et al., 2022)). The variance of local and global gradients with perturbation can be bounded as follows:*

$$\mathcal{E}_g = \|\nabla f_n(\tilde{\theta}) - \nabla f(\tilde{\theta})\|^2 \leq 3\sigma_g^2 + 6L^2\delta^2. \tag{13}$$

**Lemma 2** *(Bounded $\mathcal{E}_\epsilon$ (Qu et al., 2022)). Suppose our functions satisfies Assumptions 1-2. Then, the updates for any learning rate satisfying $\eta_l \leq \frac{1}{4TL}$ have the drift due to perturbation:*

$$\mathcal{E}_\epsilon = \mathbb{E}[\|\epsilon_{n,t} - \epsilon\|^2] \leq 2T^2L^2\eta_l^2\delta^2, \tag{14}$$

*where*

$$\epsilon_{n,t} = \delta \frac{\nabla f_n(\theta_{n,t}, \xi_n)}{\|\nabla f_n(\theta_{n,t}, \xi_n)\|}, \quad \epsilon = \delta \frac{\nabla f(\theta)}{\|\nabla f(\theta)\|}.$$

Lemma 1 bounded the variance of local and global gradients with perturbation, and it is greater than the variance of local and global gradients which mainly depends on the data-heterogeneity level.

**Lemma 3** *(Bounded model deviation). Let all assumptions hold, the deviation of the local submodel and global model with perturbation can be bounded,*

$$\frac{1}{T} \sum_{t=1}^{T} \mathbb{E}\|\tilde{\theta}_{q,n,t-1} - \tilde{\theta}_q\|\|^2 \leq 4\eta_l^2 T L^2 \delta^2 \sigma_l^2 + 32\eta_l^2 L^2 T^2 \mathcal{E}_\epsilon + 32\eta_l^2 T^2 \mathcal{E}_g$$

$$+ 4l^2 \mathbb{E}\|\theta_q\|^2 + 32\eta_l^2 T^2 \sum_{i \in \mathcal{K}_q} \mathbb{E}\|\nabla f^i(\tilde{\theta}_q)\|^2].$$

**Theorem 1** *Let all assumptions hold, suppose that the learning rates satisfy these conditions,*

$$\begin{cases} 8\eta_l^2 L^2 T^2 \leq 1 \Rightarrow \eta_l \leq \frac{1}{8LT} \\ 32\eta_l^2 T^2 \frac{N}{\mathcal{C}^*} L^2 \leq \frac{1}{16} \Rightarrow \eta_l \leq \frac{\sqrt{\mathcal{C}^*}}{16TL\sqrt{N}} \\ 96L^3\eta_l^3\eta_g T^3 \frac{N}{\mathcal{C}^*} \leq \frac{1}{16} \Rightarrow \eta_g \leq \frac{2\sqrt{N}}{\sqrt{\mathcal{C}^*}} \\ 3L\eta_l\eta_g T \leq \frac{1}{16} \Rightarrow \eta_l\eta_g \leq \frac{1}{48TL} \end{cases}$$

*Then for all $Q \geq 1$, we have*

$$\frac{1}{Q} \sum_{q=1}^{Q} \sum_{i \in \mathcal{K}_q} \mathbb{E}\|\nabla f^i(\theta_q)\|^2 \leq \frac{16\mathbb{E}[f(\theta_1)]}{T\eta_l\eta_g Q} + 64l^2(L^2 \frac{N}{\mathcal{C}^*} + 3L^3\eta_g\eta_l \frac{NT}{\mathcal{C}^*}) \frac{1}{Q} \sum_{i=1}^{Q} \mathbb{E}\|\theta_q\|^2$$

$$+ (2L^2 \frac{N}{\mathcal{C}^*} + 6L^3\eta_g\eta_l \frac{NT}{\mathcal{C}^*})(2\eta_l^2 T L^2 \delta^2 \sigma_l^2 + 16\eta_l^2 L^2 T^2 \mathcal{E}_\epsilon + 16\eta_l^2 T^2 \mathcal{E}_g)$$

$$+ (\frac{N}{\mathcal{C}^*} + 6L\eta_g\eta_l \frac{NT}{\mathcal{C}^*})\mathcal{E}_g + 3L\eta_g\eta_l \frac{N}{\mathcal{C}^*} L^2 \delta^2 \sigma_l^2.$$

The proof of the theorem can be found in the Appendix C.

**Corollary 1** *Let all assumptions hold, supposing that the step size $\eta_l = \frac{1}{\sqrt{Q}}, \eta_g = \frac{\sqrt{\mathcal{C}^*}}{\sqrt{T}}$, when the constant $C > 0$ exists, and perturbation radius $\delta$ proportional to the learning rate, e.g., $\delta = \frac{1}{\sqrt{Q}}$, the convergence rate can be expressed as follows:*

$$\frac{1}{Q} \sum_{q=1}^{Q} \sum_{i \in \mathcal{K}_q} \mathbb{E} \|\nabla f^i(\theta_q)\|^2 \leq \mathcal{O}(\frac{A_0}{\sqrt{QTC^*}} + \frac{l^2 B_0}{\mathcal{C}^*} + \frac{\sigma_g^2}{\mathcal{C}^*} + \frac{\sigma_l^2}{TQ} + \frac{1}{\sqrt{TQC^*}}),$$

*where $A_0 = \mathbb{E}[f(\theta_1)], B_0 = \frac{1}{Q} \sum_{i=1}^{Q} \mathbb{E}[f(\theta_q)]$.*

**Remark 1** *Corollary 1 indicates that when $Q$ is sufficiently large, the term $O(\frac{1}{\sqrt{QTC^*}})$ will dominate the convergence rate and the convergence increases when we properly choose the learning rate $\eta_l$ and $\eta_g$, where $\mathcal{C}^*$ is the minimum covering rate. Here we omit the higher order terms, details can be found in Appendix C. When $\mathcal{C}^* = N$, that means all the clients train full model in every communication round, the convergence rate of SubDisMO achieves to $O(\frac{1}{\sqrt{QTN}})$, which match the best convergence rate in existing general non-convex FL studies that totally train full model, such as FedAvg (Yang et al., 2021) and FedSAM (Qu et al., 2022). When $\delta = 0$, the last term is vanish and the convergence rate achieves to $O(\frac{1}{\sqrt{QTC^*}})$, which is identical to RAM-Fed (Wang et al., 2023). When $\mathcal{C}^* = 1$ and $\delta = 0$, SubDisMO achieves same convergence rate $\mathcal{O}(\frac{1}{\sqrt{QT}})$ as OAP (Zhou et al., 2023b). And these algorithms can be seen the special cases of our algorithm.*

**Remark 2** *(Impact of different key factors). Here we analyze how key factors impact the convergence of our proposed algorithm:*

- *Impact of noise induced from mask $l$. As the second term in the Corollary 1 shows, we introduce an extra term that causing the submodel mask strategy, which is proportional to the noise $l$. The smaller $l$ is, the faster the convergence rate is. And according to existing model adaptive pruning works (Ma et al., 2021), which focused on mask the insignificant parameter, it's definitely that $l^2$ is indeed small. Although clients in our algorhrim are adaptive generate submodel according to the resource, the assumption is also established. Otherwise, this term is also controlled by $\mathcal{C}^*$.*

- *Impact of data heterogeneity $\sigma_g$. As we described, the data distribution is always heterogeneous in real-world setting. Corollary 1 demonstrates that data heterogeneity is a key factor in affecting convergence. The larger $\sigma_g$ denotes the higher data heterogeneity, which can slow the convergence rate. When degenerated to iid case s.t. $\sigma_g = 0$, this term becomes zero, which is faster than existing convergence rate.*

- *Impact of trained parameters $|\mathcal{K}_q|$. We innovative analyze the convergence rate of our algorithm that separates the trained model parameters $\mathcal{K}_q$ and untrained model parameters $\mathcal{S} - \mathcal{K}_q$ in communication round $q$, so we give a rigour bound of the averaged gradient of the trained parameters. It is intuitive that the larger $\mathcal{K}_q$, the more parameters can be trained, the more the gradient of the parameters can be bounded, which benefits model convergence.*

**Remark 3** *The additional term $\mathcal{O}(\frac{1}{Q^3 \mathcal{C}^*})$ comes from the extra local updates due to the perturbation via Eq. 5. And the local updates drift we've analyzed in Lemma 1. However, it can be neglected owing to its higher order. Thus, we improve the generalization of the model through a little computation but without slowing down the convergence rate.*

## 4.2 GENERALIZATION BOUNDS OF SUBDISMO

**Margin Loss.** First, in order to bound the generalization error of SubDisMO, for margin $\gamma > 0$, we define the expected margin loss as follows.

$$\mathcal{L}_\gamma(f(\theta)) := \frac{1}{N} \sum_{i=1}^{N} \mathbb{P}_i \left[ f_i(\theta \odot m_i + \epsilon_i, X)[y] \leq \gamma + \max_{z \neq y} f_i(\theta \odot m_i + \epsilon_i, X)[z] \right].$$

Here, $f_i(\theta \odot m_i + \epsilon_i, X)$ is the local loss function for client $i$ as defined in (2), $(X, y)$ is a sample from the local distribution of client $i$. $f_i(\theta \odot m_i + \epsilon_i, X)[z]$ is the output of the last softmax layer of the training neural network for label $z$. Let $\hat{\mathcal{L}}_\gamma(f(\theta))$ be the empirical estimate of the above expected margin loss on the training dataset with $d$ samples.

Therefore, we aim to give bounds on the difference between the expected risk and the empirical margin-based error. First, we use the following lemma that gives a margin-based generalization bound derived from the PAC-Bayesian bound.

**Lemma 4** *(Bounded margin-based generalization (Neyshabur et al., 2018)). Let $f(\theta)$ be any predictor with parameter $\theta$, and prior $P$ be any distribution on the parameter $\theta$ that is independent of the training data. Then, for any $\gamma, \zeta, d > 0$, with probability $1 - \zeta$ over training set $\mathcal{D}$ of size $d$, for any $\theta$, and any perturbation $\epsilon$ s.t. $\mathbb{P}_\epsilon[\max_X |f(\theta \odot m + \epsilon) - f(\theta)|_\infty \leq \frac{\gamma}{4}] \geq \frac{1}{2}$, we have:*

$$\mathcal{L}(f(\theta)) \leq \hat{\mathcal{L}}_\gamma(f(\theta)) + 4\sqrt{\frac{KL(\theta \odot m + \epsilon \parallel P) + \ln\frac{6d}{\zeta}}{d - 1}}, \tag{15}$$

*where $KL(\cdot \parallel P)$ is the KL-divergence.*

In order to bound the change in the output of the network when only partial network are trained and perturbed, referring to Neyshabur et al. (2018), we give the following lemma in terms of the spectral norm of the layers.

**Lemma 5** *(Perturbed submodel Bound). Let the norm of input $X$ be bounded by $A$. For any $A, l > 0$, let $f(\theta)$ be a $r$-layer neural network with ReLU activations, and $j$-th layer has $h_j$ units. Then for any $\theta$, $\theta = vec(\{\Theta_j\}_{j=1}^r)$, and any perturbation $\epsilon = vec(\{\epsilon_j\}_{j=1}^r)$, s.t. $\|\epsilon_j\|_2 \leq \frac{1}{r}\|\theta_j\|_2$, $s_j$ denotes the remaining rate in layer $j$, $0 < s_j \leq 1$. The change of the network can be bound as follows:*

$$|f(\theta \odot m + \epsilon) - f(\theta)|_2 \leq A\prod_{j=1}^r(s_j + \frac{1}{r})\prod_{j=1}^r\|\theta_j\|_2\sum_{j=1}^r\frac{\|\epsilon_j\|_2}{\|\theta_j\|_2}. \tag{16}$$

The proof of this lemma can be found in the Appendix D. Then, we use the above two lemmas to derive the following generalization guarantee.

**Theorem 2** *(Generalization bounds of SubDisMO). For any $A, r, h_j > 0$, let $\tilde{h} = \max s_j h_j$ be an upper bound on the unit number in each layer of submodel. Assume for constant $M \geq 1$ any layer $\theta_j$ satisfies $\frac{1}{M} \leq \frac{\|\theta_j\|_2}{\beta} \leq M$, where $\beta := (\prod_{j=1}^r\|\theta_j\|_2)^{1/r}$. Then for any $\gamma, \zeta > 0$, with probability $1 - \zeta$ over training set $\mathcal{D}$ of size $d$, for any parameter $\theta$, with $\epsilon \sim \mathcal{N}(0, \sigma^2 I)$, s.t. $\sigma \leq \frac{\gamma}{16\prod_{j=1}^r(s_j + \frac{1}{r})rA\tilde{\beta}^{r-1}\sqrt{\tilde{h}\ln(4r\tilde{h})}}$, $\tilde{\beta}$ is an approximation to $\beta$, we have:*

$$\mathcal{L}(f(\theta)) \leq \hat{\mathcal{L}}_\gamma(f(\theta)) + \mathcal{O}(\sqrt{\frac{\prod_{j=1}^r(s_j + \frac{1}{r})^2 r^2 A^2 \ln(r\tilde{h})S(\theta) + r\ln\frac{Nrd\log M}{\zeta}}{d\gamma^2}}), \tag{17}$$

*where $S(\theta) = \prod_{j=1}^r\|\theta_j\|_2^2\sum_{i=j}^r\frac{s_j\|\theta_j\|_F^2}{\|\theta_j\|_2^2}$, $\|\theta_j\|_F$ is the Frobenius norm.*

The proof of the theorem can be found in the Appendix D.

**Remark 4** *Theorem 2 gives an asymptotic bound on the generalization risk of SubDisMO for general neural network. Compared to the traditional PAC-Bayesian bound of the perturbed model (Neyshabur et al., 2018; Qu et al., 2022; Chatterji et al., 2020), it introduces the remaining rate in each layer to the bound. When each client trains the full model without the mask, s.t. $s_j = 1$, thus $\prod_{j=1}^l(s_j + \frac{1}{l}) = (1 + \frac{1}{l})^l \leq e$, as $1 + x \leq e^x$, for all $x$, the generalization bound is similar to the asymptotic bound as shown in (Qu et al., 2022). It means that we not only give bounds on the difference between the empirical error and the expected margin-based error, but also give a tighter bound compared to the existing work, where each client only trains a submodel. And we also present how to properly choose the $\epsilon \sim \mathcal{N}(0, \sigma^2 I)$ to be the perturbation so that we can guarantee the generalization of SubDisMO.*

Table 1: Test accuracy (%) and standard deviation on CIFAR-10 & CIFAR-100 datasets under different data distributions.

| | Algorithm | ViT-Small/CIFAR-10 | | | ViT/CIFAR-100 | | |
| | | $\mu = 0.5$ | $\mu = 1.0$ | IID | $\mu = 0.5$ | $\mu = 1.0$ | IID |
| | | Acc.(Dev.) | Acc.(Dev.) | Acc.(Dev.) | Acc.(Dev.) | Acc.(Dev.) | Acc.(Dev.) |
|---|---|---|---|---|---|---|---|
| Full | FedAvg | 56.44(5.28) | 55.70(2.96) | 58.75(1.65) | 31.60(2.50) | 30.95(2.90) | 32.17(0.95) |
| | FedSAM | 56.02(4.80) | 57.03(2.62) | 59.01(1.58) | 31.73(2.38) | 32.11(2.68) | 33.36(1.09) |
| Sub. | IST.O | 36.82(16.67) | 38.06(10.27) | 45.81(2.03) | 16.35(7.17) | 17.15(5.54) | 19.99(0.48) |
| | IST.P | 33.66(14.35) | 37.96(9.08) | 45.68(1.47) | 15.67(6.59) | 17.59(4.86) | 19.98(0.43) |
| | IST.S | 30.73(14.36) | 33.78(8.72) | 41.26(1.52) | 15.70(6.82) | 16.91(5.29) | 18.02(2.91) |
| | IST.A | 39.02(12.13) | 41.26(8.48) | 47.59(1.49) | 18.17(5.60) | 19.02(2.81) | 21.57(1.57) |
| | OAP.O | 45.53(12.13) | 48.29(9.29) | 53.69(1.52) | 21.91(6.92) | 26.09(6.97) | 27.69(1.65) |
| | OAP.P | 41.55(12.51) | 45.99(7.18) | 53.70(1.56) | 20.81(6.35) | 24.24(4.10) | 26.14(1.51) |
| | OAP.S | 32.12(10.63) | 41.55(7.17) | 46.15(1.67) | 21.34(2.46) | 22.10(3.10) | 23.33(1.08) |
| | OAP.A | 37.27(9.65) | 42.72(7.16) | 47.04(1.34) | 21.56(3.70) | 23.40(3.68) | 26.16(1.57) |
| | PruneFL.O | 44.87(14.90) | 48.20(5.16) | 53.04(1.56) | 20.36(6.90) | 22.35(5.80) | 22.09(1.36) |
| | PruneFL.P | 44.02(12.64) | 49.71(4.25) | 52.35(1.33) | 15.75(6.22) | 17.89(5.44) | 20.43(1.46) |
| | PruneFL.S | 37.22(11.56) | 47.11(5.32) | 43.20(1.50) | 17.19(3.59) | 20.17(3.22) | 20.89(1.01) |
| | PruneFL.A | 39.32(14.60) | 41.31(14.59) | 52.78(1.23) | 15.30(5.25) | 16.54(3.99) | 19.96(0.88) |
| | FedRolex.O | 40.07(12.40) | 44.84(4.75) | 45.46(2.08) | 20.69(2.93) | 21.73(2.57) | 22.34(1.46) |
| | FedRolex.P | 41.12(11.87) | 45.27(6.16) | 49.72(1.51) | 21.12(3.59) | 21.74(4.34) | 24.43(1.28) |
| | FedRolex.S | 35.12(11.67) | 40.60(7.08) | 45.53(1.46) | 19.24(4.97) | 20.78(4.90) | 23.42(1.44) |
| | FedRolex.A | 37.58(10.42) | 43.41(6.38) | 47.23(1.33) | 20.28(5.72) | 22.66(5.72) | 24.91(1.61) |
| | RAM-Fed | 43.31(11.49) | 50.19(4.16) | 53.33(1.42) | 20.42(5.19) | 23.25(4.77) | 24.52(0.84) |
| | **SubDisMO** | **48.50**(8.47) | **51.23**(4.77) | **55.99**(1.85) | **23.17**(5.60) | **25.43**(4.56) | **28.24**(1.27) |

## 5 EXPERIMENTS

In this section, we focus on the generalization and effectiveness of our proposed *SubDisMO* compared with some federated learning algorithms combined with resource-limited training paradigms. Moreover, we also explore the effect of two key factors in our algorithm including minimum covering number $\mathcal{C}^*$ and the upper bound of the perturbation $\delta$. Due to the space limitation, further scalability analysis and computation analysis are represented in Appendix E.

### 5.1 EXPERIMENTAL SETUP

**Datasets and models.** We compare the performance of baselines on two traditional image classification datasets: *CIFAR-10* (Krizhevsky et al., 2009) with ViT-small and *CIFAR-100* (Krizhevsky et al., 2009) with ViT. Both models are based on Transformer architecture, with detailed settings are shown in Table 4 in Appendix E. To show the effect of model architecutures, we additionally add the experiments using ResNet18 on *CIFAR-10*, results are shown in Table 5 in Appendix E. We use two settings to simulate heterogeneous data distributions among 10 clients. In the IID setting, each client has the same number of samples from all classes. In the non-IID setting, data heterogeneity levels are determined by the Dirichlet distribution Dir($\mu$) (Hsu et al., 2019), with $\mu = 0.5$ simulating high heterogeneity and $\mu = 1.0$ representing lower heterogeneity. In order to test the generalization of the global model, we also split the test dataset according to the same distribution among the 10 clients.

**Baselines.** We compare our proposed *SubDisMO* with several combinations of resource-limited distributed learning paradigms and kinds of aggregation algorithms. We choose *IST* (Yuan et al., 2022), *PruneFL* (Jiang et al., 2022), *OAP* (Zhou et al., 2023b), and *FedRolex* (Alam et al., 2022) as the basic resource-limited distributed paradigms. For the aggregation algorithms, we use *FedAvg* (McMahan et al., 2017), *FedProx* (Li et al., 2020), *SCAFFOLD* (Karimireddy et al., 2020), and *FedAdam* (Reddi et al., 2020), denoted by *O, P, S, A* for simplicity. We additionally compared with *RAM-Fed* (Wang

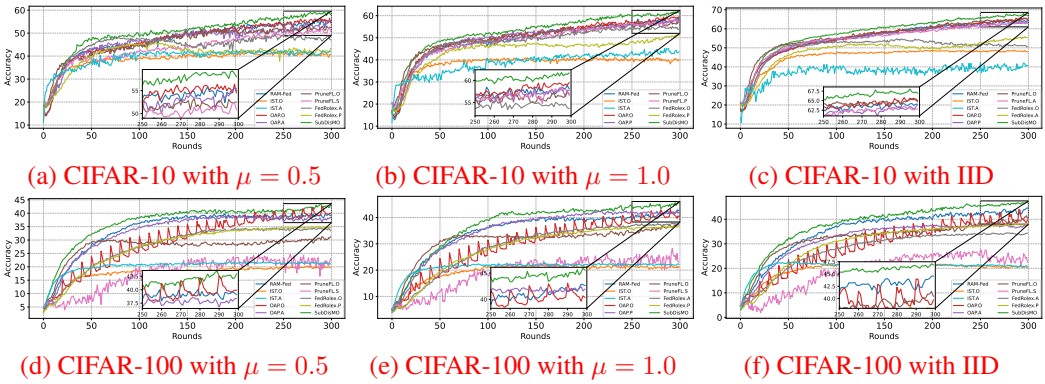

(a) CIFAR-10 with $\mu = 0.5$     (b) CIFAR-10 with $\mu = 1.0$     (c) CIFAR-10 with IID

(d) CIFAR-100 with $\mu = 0.5$     (e) CIFAR-100 with $\mu = 1.0$     (f) CIFAR-100 with IID

Figure 2: Training process of different learning paradigms.

et al., 2023), which focuses on solving non-iid data under resource-limited settings. Besides, we use the *FedAvg* and *FedSAM* on full model to show the best performance without model pruning.

**Submodel setting.** In each communication round, we randomly split the full model into four submodels $\theta_1, \theta_2, \theta_3, \theta_4$ without overlap, so that each submodel contains 25% of the parameters. To address the model heterogeneity, 50% of the clients with low resources arbitrarily train 1/4 of the parameters, while the remaining 50% of the clients train 1/2 of the parameters. Low-resource clients randomly choose one submodel to train, whereas the remaining clients choose two parts (e.g., $\theta_1, \theta_2$) to form local submodel. Specially, for the *IST* design, the full model is divided into 10 equal submodels, with each client training one part. In *PruneFL*, clients only train the most important submodel, meaning that only portion of the parameters can be trained.

## 5.2 PERFORMANCE EVALUATION

The main results of *SubDisMO* compared to all the baselines are shown in Table 1. We report the average test accuracy and standard deviation across the clients' test datsets.

**Performance compared with baselines.** Overall, our proposed *SubDisMO* outperforms other baselines in terms of average accuracy and maintains lower deviation, except for FedAvg and FedSAM with the full model. Compared to the second-best results, *SubDisMO* improves accuracy by 1.52%-2.97% on *CIFAR-10* and 0.55%-1.26% on *CIFAR-100*, demonstrating the efficiency of our proposed method. Notably, our method achieves lower standard deviations while ensuring higher accuracy, indicating the excellent generalization of our *SubDisMO*. For example, *OAP.A* achieves the lowest deviation on Dir($\mu = 1.0$) for *CIFAR-10*, its average accuracy is significantly lower than ours. This means that although *OAP.A* performs consistently across different clients, its overall performance is subpar. In contrast, *SubDisMO* outperforms this baseline by 6.11% with a lower deviation. The convergence process of different learning paradigms is depicted in Figure 3. Considering the readable, we only choose the top-2 methods for each resource-limited distributed paradigm.

**Impact of non-iid data.** With the increment of the data heterogeneity level, that is $\mu$ becomes smaller, the average accuracy of all methods decreases. However, our method still outperforms all baselines. Additionally, the deviation among clients increases with higher $\mu$, even for the federated full model training baselines, indicating that data distribution impacts the generalization of the global model. Nevertheless, our method decreases slightly than other baselines, confirming its effectiveness in mitigating the adverse effects of data heterogeneity. This demonstrates that our method not only maintains higher average accuracy across varying levels of data heterogeneity but also reduces the variance in performance among clients. Specifically, *RAM-Fed*, another resource-adaptive learning paradigm focused on the non-iid data, is outperformed by our method, further showcasing the superior generalization capabilities of *SubDisMO*.

**Loss landscape visualization.** As previously mentioned, *arbitrary submodel sharpness* negatively impacts the generalization of the global model. Thus we visualize the the loss landscape of the global model both in *RAM-Fed* and our *SubDisMO* trained on *CIFAR-10* under Dir($\mu$=0.5) following the plotting algorithm in literature (Li et al., 2018). As shown in Figure 3, we can observe that the

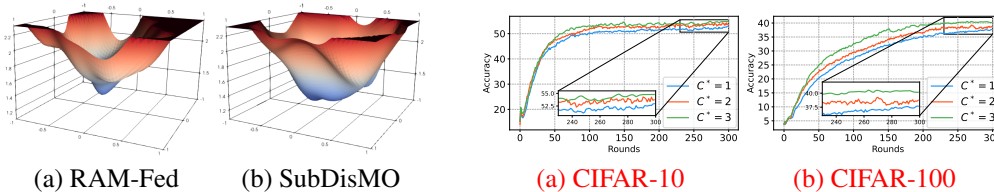

(a) RAM-Fed      (b) SubDisMO          (a) CIFAR-10          (b) CIFAR-100

Figure 3: Visualization of loss landscape.          Figure 4: The impact of different $\mathcal{C}^*$.

*SubDisMO* can mitigate sharpness and make the loss landscape flatter, despite each client only trains a submodel, which indicates that our method improves the generalization significantly.

### 5.3 IMPACT OF KEY FACTORS

To explore the influence of different factors, we conduct experiments on two key factors of *SubDisMO*.

**Impact of $\delta$.** In order to investigate the impact of the perturbation radius $\delta$ on *SubDisMO*, we fix other settings and choose different value of $\delta$ to run the algorithm within the Dir($\mu = 1.0$) distribution. The convergence results and final test results are shown in Table 2. We see that when $\delta \leq 0.1$ for *CIFAR-10*, as $\delta$ increases, the average test accuracy improves and the deviations among clients decrease, which shows the introduction of $\delta$ enhances both generality and performance. For *CIFAR-100*, when $\delta \leq 0.1$, the average test accuracy and deviation are almost no change, and when $\delta \geq 0.1$, as $\delta$ increases, the average test accuracy decreases while deviation gets small. But for *CIFAR-10*, when $\delta$ continues

Table 2: Impact of hyperparameter $\delta$ on CIFAR-10 & CIFAR-100 datasets.

| Algorithm | ViT-Small/CIFAR-10 | | ViT/CIFAR-100 | |
|---|---|---|---|---|
| | Acc. | Dev. | Acc. | Dev. |
| $\delta = 0.01$ | 49.73 | 5.84 | 25.55 | 4.62 |
| $\delta = 0.05$ | 50.66 | 5.76 | 25.51 | 4.53 |
| $\delta = 0.08$ | 50.99 | 5.48 | 25.41 | 4.5 |
| $\delta = 0.1$ | 51.23 | 4.77 | 25.43 | 4.56 |
| $\delta = 0.15$ | 50.30 | 5.31 | 24.59 | 4.26 |
| $\delta = 0.2$ | 50.58 | 6.06 | 23.51 | 3.83 |
| $\delta = 0.3$ | 50.48 | 6.07 | 22.85 | 3.39 |

to grow, the performance of *SubDisMO* declines in both accuracy and generality. This decline is due to excessive perturbations causing model parameters to deviate from the local minima, which adversely affects the model.

**Impact of $\mathcal{C}^*$.** We manually set the submodel that each client trains to explore the impact of the minimum covering rate $\mathcal{C}^*$. Considering a heterogeneous resource setting, $\mathcal{C}^*$ is set to $1, 2, 3$, ensuring all parameters are trained. The results are shown in Figure 4. We observe that when all parameters are covered, the more frequently the parameters are trained, the higher the accuracy. When $\mathcal{C}^* = 3$, the test accuracy is the best. Additionally, considering the convergence rate, we find that a larger $\mathcal{C}^*$ leads to faster convergence, consistent with our theoretical analysis.

## 6 CONCLUSION

Distributed minimax optimization faces challenges when devices are constrained by limited computing and communication resources. In this work, we designed a resource-aware algorithm, *SubDisMO*, under distributed minimax optimization to address the *arbitrary submodel sharpness* caused by data heterogeneity while training perturbed submodels on resource-limited devices. We theoretically proved that *SubDisMO* can achieve asymptotically optimal convergence rate $\mathcal{O}(1/\sqrt{QT\mathcal{C}^*})$ under general non-convex distributed assumptions. Furthermore, we analyzed the impact of noise induced by masking, data heterogeneity, and partially trained parameters on the convergence rate. Otherwise, we gave a generalization bound of *SubDisMO* corresponding to the perturbation and parameter remaining rate in each layer. Extensive experiments confirmed that *SubDisMO* improves overall performance while reducing deviation among clients.

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

APPENDIX

We provide more details about our work and experiments in the appendices:

- Appendix A: the frequently used notations in this work.
- Appendix B: the preliminary lemmas used in the theoretical analysis.
- Appendix C: details proof of the convergence analysis of our proposed *SubDisMO*.
- Appendix D: details proof of the generalization bound of our proposed *SubDisMO*.
- Appendix E: the details of experiments settings and supplemental experiment results.
- Appendix F: the details of limitations and broader impacts of this work.

# A    NOTATIONS

Table 3: Frequently used notations

| Notations | Descriptions |
|---|---|
| $\|\cdot\|$ | the vector $\ell_2$ norm or the matrix spectral norm depending on the argument |
| $\mathcal{S}$ | the set of all trained parameters |
| $\mathcal{K}_q$ | the trained parameters set in round $q$ |
| $|\mathcal{K}_q|$ | the number of trained parameters in round $q$ |
| $N_q^i$ | the set of clients training parameter $i$ in round $q$ |
| $\mathcal{C}_q^i$ | $\mathcal{C}_q^i = |N_q^i|$ the number of clients in $N_q^i$ |
| $\mathcal{C}^*$ | minimum covering number: $\mathcal{C}^* = \min_{q,i} \mathcal{C}_q^i, i \in \mathcal{K}_q, \forall q$ |
| $\Delta_q^i$ | the accumulated updates for parameter $i$ of global model in round $q$ |
| $\Delta_{q,n}$ | the accumulated local updates from client $n$ on itself submodel in round $q$ |
| $\Delta_{q,n}^i$ | the accumulated local updates from client $n$ on parameter $i$ in round $q$ |
| $m_{q,n}$ | the mask of client $n$ in round $q$ |
| $\theta_q$ | the global model in round $q$ |
| $\theta_q^i$ | the parameter $i$ of global model in round $q$ |
| $\tilde{\theta}_q$ | the perturbed global model in round $q$ |
| $\tilde{\theta}_q^i$ | the perturbed parameter $i$ of global model in round $q$ |
| $\epsilon_i$ | the perturbation in $i$-th client |
| $\delta$ | the radius of perturbation |
| $f_n(\theta, \xi_n)$ | the loss function for client $n$ |
| $\nabla f_n(\theta)$ | $\mathbb{E}_{\xi_n \sim D_n} \nabla f_n(\theta, \xi_n)$ |
| $\eta_l$ | the learning rate of clients |
| $\eta_g$ | the learning rate of server |
| $\mathcal{L}(f(\theta))$ | the expected loss |
| $\tilde{\mathcal{L}}(f(\theta))$ | the empirical loss |
| $\gamma$ | margin |
| $d$ | local training data samples |
| $P$ | prior distribution |
| $A$ | bound of the norm of input $X$ |
| $r$ | the layer number of neural network |
| $h_j$ | the units number of $j$-th layer |
| $s_j$ | remaining rate in $j$-th layer |
| $\tilde{h} = \max s_j h_j$ | upper bound on the unit number in each layer of submodel |
| $\beta$ | $(\prod_{j=1}^r \|\theta_j\|_2)^{1/r}$, geometric mean of the $\theta$'s spectral norm across all layers |
| $\tilde{\beta}$ | appropriation of $\beta$ |

## B PRELIMINARY LEMMAS

In order to analyze the convergence rate of our proposed *FedMKD*, we firstly state some preliminary lemmas as follows:

**Lemma 6** *(Jensen's inequality). For any convex function $h$ and any variable $x_1, \ldots, x_n$ we have*

$$h(\frac{1}{n}\sum_{i=1}^{n}x_i) \leq \frac{1}{n}\sum_{i=1}^{n}h(x_i). \tag{18}$$

*Especially, when $h(x) = \|x\|^2$, we can get*

$$\|\frac{1}{n}\sum_{i=1}^{n}x_i\|^2 \leq \frac{1}{n}\sum_{i=1}^{n}\|x_i\|^2. \tag{19}$$

**Lemma 7** *For random variable $x_1, \ldots, x_n$ we have*

$$\mathbb{E}[\|x_1 + \cdots + x_n\|^2] \leq n\mathbb{E}[\|x_1\|^2 + \cdots + \|x_n\|^2]. \tag{20}$$

**Lemma 8** *For independent random variables $x_1, \ldots, x_n$ whose mean is 0, we have*

$$\mathbb{E}[\|x_1 + \cdots + x_n\|^2] = \mathbb{E}[\|x_1\|^2 + \cdots + \|x_n\|^2]. \tag{21}$$

***Proof of Lemma 1.***

$$
\begin{aligned}
\|\nabla f_n(\tilde{\theta}) - \nabla f(\tilde{\theta})\|^2 &= \|\nabla f_n(\theta + \epsilon_n) - \nabla f(\theta + \epsilon)\|^2 \\
&= \|\nabla f_n(\theta + \epsilon_n) - \nabla f_n(\theta) + \nabla f_n(\theta) - \nabla f(\theta) + \nabla f(\theta) - \nabla f(\theta + \epsilon)\|^2 \\
&\leq 3\|\nabla f_n(\theta + \epsilon_n) - \nabla f_n(\theta)\|^2 + 3\|\nabla f_n(\theta) - \nabla f(\theta)\|^2 + 3\|\nabla f(\theta) - \nabla f(\theta + \epsilon)\|^2 \\
&\leq 3\sigma_g^2 + 6L^2\delta^2,
\end{aligned}
$$

## C CONVERGENCE ANALYSIS

***Proof of Theorem 1.*** Let us start the proof of the global model generated by semi-asynchronous aggregation strategy from $L$-Lipschitzian Condition:

$$\mathbb{E}[f(\theta_{q+1})] = \mathbb{E}[f(\tilde{\theta}_{q+1})] \leq f(\tilde{\theta}_q) + \underbrace{\mathbb{E}[\langle \nabla f(\tilde{\theta}_q), \tilde{\theta}_{q+1} - \tilde{\theta}_q\rangle]}_{U_1} + \underbrace{\frac{L}{2}\mathbb{E}\|\tilde{\theta}_{q+1} - \tilde{\theta}_q\|^2}_{U_2}$$

To bound $U_1$:

$$
\begin{aligned}
&\mathbb{E}[\langle \nabla f(\tilde{\theta}_q), \tilde{\theta}_{q+1} - \tilde{\theta}_q\rangle] \\
&= \sum_{i\in\mathcal{K}_q}\mathbb{E}[\langle \nabla f(\tilde{\theta}_q), \tilde{\theta}_{q+1} - \tilde{\theta}_q\rangle] + \sum_{i\in\mathcal{S}-\mathcal{K}_q}\mathbb{E}[\langle \nabla f(\tilde{\theta}_q), \tilde{\theta}_{q+1} - \tilde{\theta}_q\rangle] \\
&= \sum_{i\in\mathcal{K}_q}\mathbb{E}[\langle \nabla f(\tilde{\theta}_q), \tilde{\theta}_{q+1} - \tilde{\theta}_q\rangle] + \sum_{i\in\mathcal{S}-\mathcal{K}_q}\mathbb{E}[\langle \nabla f(\tilde{\theta}_q), \mathbf{0}\rangle] \\
&= \sum_{i\in\mathcal{K}_q}\mathbb{E}[\langle \nabla f(\tilde{\theta}_q), \tilde{\theta}_{q+1} - \tilde{\theta}_q\rangle] \\
&= \sum_{i\in\mathcal{K}_q}\mathbb{E}[\langle \nabla f^i(\tilde{\theta}_q), -\eta_g\Delta_q^i)\rangle] + \sum_{i\in\mathcal{K}_q}\mathbb{E}[\langle \nabla f^i(\tilde{\theta}_q), \epsilon_{q+1} - \epsilon_q\rangle] \\
&\leq \sum_{i\in\mathcal{K}_q}\mathbb{E}[\langle \nabla f^i(\tilde{\theta}_q), -\eta_g(\frac{1}{\mathcal{C}_q^i}\sum_{n\in N_q^i}(\theta_{q,n,0}^i - \theta_{q,n,T}^i))\rangle] + \frac{\eta_l\eta_g T}{4}\sum_{i\in\mathcal{K}_q}\mathbb{E}\|\nabla f^i(\tilde{\theta}_q)\|^2 + \frac{1}{\eta_l\eta_g T}\sum_{i\in\mathcal{K}_q}\mathbb{E}\|\epsilon_{q+1} - \epsilon_q\|^2 \\
&\leq \sum_{i\in\mathcal{K}_q}\mathbb{E}[\langle \nabla f^i(\tilde{\theta}_q), -\frac{\eta_g}{\mathcal{C}_q^i}\sum_{n\in N_q^i}(\theta_{q,n,0} - (\theta_{q,n,0} - \sum_{t=1}^{T}\eta_l\nabla f_n(\tilde{\theta}_{q,n,t-1}, \xi_{n,t-1})\odot m_{q,n}))^i)\rangle] + \frac{\eta_l\eta_g T}{4}\sum_{i\in\mathcal{K}_q}\mathbb{E}\|\nabla f^i(\tilde{\theta}_q)\|^2 + \frac{1}{\eta_l\eta_g T}\delta^2 \\
&= \sum_{i\in\mathcal{K}_q}\mathbb{E}[\langle \nabla f^i(\tilde{\theta}_q), -\frac{\eta_g}{\mathcal{C}_q^i}\sum_{n\in N_q^i}\sum_{t=1}^{T}\eta_l\nabla f_n^i(\tilde{\theta}_{q,n,t-1}, \xi_{n,t-1})\rangle] + \frac{\eta_l\eta_g T}{4}\sum_{i\in\mathcal{K}_q}\mathbb{E}\|\nabla f^i(\tilde{\theta}_q)\|^2 + \frac{1}{\eta_l\eta_g T}\delta^2
\end{aligned}
$$

$$= \sum_{i \in \mathcal{K}_q} \mathbb{E}[\langle \nabla f^i(\tilde{\theta}_q), -\frac{\eta_g}{\mathcal{C}_q^i} \sum_{n \in N_q^i} \sum_{t=1}^{T} \eta_l \nabla f_n^i(\tilde{\theta}_{q,n,t-1}) \rangle] + \frac{\eta_l \eta_g T}{4} \sum_{i \in \mathcal{K}_q} \mathbb{E}\|\nabla f^i(\tilde{\theta}_q)\|^2 + \frac{1}{\eta_l \eta_g T} \delta^2$$

$$= \sum_{i \in \mathcal{K}_q} \mathbb{E}[\langle \nabla f^i(\tilde{\theta}_q), -\frac{\eta_g}{\mathcal{C}_q^i} \sum_{n \in N_q^i} \sum_{t=1}^{T} \eta_l [\nabla f_n^i(\tilde{\theta}_{q,n,t-1}) - \nabla f^i(\tilde{\theta}_q) + \nabla f^i(\tilde{\theta}_q)] \rangle] + \frac{\eta_l \eta_g T}{4} \sum_{i \in \mathcal{K}_q} \mathbb{E}\|\nabla f^i(\tilde{\theta}_q)\|^2 + \frac{1}{\eta_l \eta_g T} \delta^2$$

$$= \underbrace{- \sum_{i \in \mathcal{K}_q} T \eta_g \eta_l \mathbb{E}[\langle \nabla f^i(\tilde{\theta}_q), \nabla f^i(\tilde{\theta}_q) \rangle]}_{U_3} + \underbrace{\sum_{i \in \mathcal{K}_q} \mathbb{E}[\langle \nabla f^i(\tilde{\theta}_q), -\frac{\eta_g \eta_l}{\mathcal{C}_q^i} \sum_{n \in N_q^i} \sum_{t=1}^{T} [\nabla f_n^i(\tilde{\theta}_{q,n,t-1}) - \nabla f^i(\tilde{\theta}_q)] \rangle]}_{U_4}$$

$$+ \frac{\eta_l \eta_g T}{4} \sum_{i \in \mathcal{K}_q} \mathbb{E}\|\nabla f^i(\tilde{\theta}_q)\|^2 + \frac{1}{\eta_l \eta_g T} \delta^2$$

To bound $U_3$:

$$- \sum_{i \in \mathcal{K}_q} T \eta_g \eta_l \mathbb{E}[\langle \nabla f^i(\tilde{\theta}_q), \nabla f^i(\tilde{\theta}_q) \rangle] = - \sum_{i \in \mathcal{K}_q} T \eta_g \eta_l \mathbb{E}\|\nabla f^i(\tilde{\theta}_q)\|^2$$

bound $U_4$:

$$\sum_{i \in \mathcal{K}_q} \mathbb{E}[\langle \nabla f^i(\tilde{\theta}_q), -\frac{\eta_g \eta_l}{\mathcal{C}_q^i} \sum_{n \in N_q^i} \sum_{t=1}^{T} [\nabla f_n^i(\tilde{\theta}_{q,n,t-1}) - \nabla f^i(\tilde{\theta}_q)] \rangle]$$

$$= \sum_{i \in \mathcal{K}_q} \eta_g \eta_l T \mathbb{E}[< \nabla f^i(\tilde{\theta}_q), -\frac{1}{T\mathcal{C}_q^i} \sum_{n \in N_q^i} \sum_{t=1}^{T} [\nabla f_n^i(\tilde{\theta}_{q,n,t-1}) - \nabla f^i(\tilde{\theta}_q)] >$$

$$\leq \frac{\eta_g \eta_l T}{2} \sum_{i \in \mathcal{K}_q} \mathbb{E}\|\nabla f^i(\tilde{\theta}_q)\|^2 + \frac{\eta_g \eta_l T}{2} \sum_{i \in \mathcal{K}_q} \mathbb{E}\|\frac{1}{T\mathcal{C}_q^i} \sum_{n \in N_q^i} \sum_{t=1}^{T} [\nabla f_n^i(\tilde{\theta}_{q,n,t-1}) - \nabla f_n^i(\tilde{\theta}_q) + \nabla f_n^i(\tilde{\theta}_q) - \nabla f^i(\tilde{\theta}_q)]\|^2$$

$$\leq \frac{\eta_g \eta_l T}{2} \sum_{i \in \mathcal{K}_q} \mathbb{E}\|\nabla f^i(\tilde{\theta}_q)\|^2 + \underbrace{\eta_g \eta_l T \sum_{i \in \mathcal{K}_q} \mathbb{E}\|\frac{1}{T\mathcal{C}_q^i} \sum_{n \in N_q^i} \sum_{t=1}^{T} [\nabla f_n^i(\tilde{\theta}_{q,n,t-1}) - \nabla f_n^i(\tilde{\theta}_q)]\|^2}_{U_5}$$

$$\underbrace{+ \eta_g \eta_l T \sum_{i \in \mathcal{K}_q} \mathbb{E}\|\frac{1}{T\mathcal{C}_q^i} \sum_{n \in N_q^i} \sum_{t=1}^{T} [\nabla f_n^i(\tilde{\theta}_q) - \nabla f^i(\tilde{\theta}_q)]\|^2}_{U_6}$$

To bound $U_5$:

$$\eta_g \eta_l T \sum_{i \in \mathcal{K}_q} \mathbb{E}\|\frac{1}{T\mathcal{C}_q^i} \sum_{n \in N_q^i} \sum_{t=1}^{T} [\nabla f_n^i(\tilde{\theta}_{q,n,t-1}) - \nabla f_n^i(\tilde{\theta}_q)]\|^2$$

$$\leq \eta_g \eta_l T \sum_{i \in \mathcal{K}_q} \frac{1}{T\mathcal{C}_q^i} \sum_{n \in N_q^i} \sum_{t=1}^{T} \mathbb{E}\|[\nabla f_n^i(\tilde{\theta}_{q,n,t-1}) - \nabla f_n^i(\tilde{\theta}_q)]\|^2$$

$$\leq \eta_g \eta_l T \frac{1}{T\mathcal{C}^*} \sum_{n=1}^{N} \sum_{t=1}^{T} \sum_{i \in \mathcal{K}_q} \mathbb{E}\|[\nabla f_n^i(\tilde{\theta}_{q,n,t-1}) - \nabla f_n^i(\tilde{\theta}_q)]\|^2$$

$$\leq \eta_g \eta_l T \frac{1}{T\mathcal{C}^*} \sum_{n=1}^{N} \sum_{t=1}^{T} \mathbb{E}\|[\nabla f_n(\tilde{\theta}_{q,n,t-1}) - \nabla f_n(\tilde{\theta}_q)]\|^2$$

$$\leq \eta_g \eta_l T \frac{1}{\mathcal{C}^*} \sum_{n=1}^{N} L^2 \frac{1}{T} \sum_{t=1}^{T} \mathbb{E}\|\tilde{\theta}_{q,n,t-1} - \tilde{\theta}_q]\|^2$$

$$\leq \eta_g \eta_l T \frac{1}{\mathcal{C}^*} \sum_{n=1}^{N} L^2 \frac{1}{T} \sum_{t=1}^{T} \mathbb{E}\|\theta_{q,n,t-1} + \epsilon_{q,n,t-1} - \theta_q - \epsilon_q]\|^2$$

$$\leq 2\eta_g\eta_l T\frac{1}{\mathcal{C}^*}\sum_{n=1}^{N}L^2\underbrace{\frac{1}{T}\sum_{t=1}^{T}\mathbb{E}\|\theta_{q,n,t-1}-\theta_q\|^2}_{U_7}+2\eta_g\eta_l T\frac{1}{\mathcal{C}^*}\sum_{n=1}^{N}L^2\frac{1}{T}\sum_{t=1}^{T}\mathbb{E}\|\epsilon_{q,n,t-1}-\epsilon_q]\|^2$$

To bound $U_7$:

$$\frac{1}{T}\sum_{t=1}^{T}\mathbb{E}\|\theta_{q,n,t-1}-\theta_q\|^2$$

$$\leq \frac{2}{T}\sum_{t=1}^{T}\mathbb{E}\|\theta_{q,n,t-1}-\theta_{q,n,0}\|^2+\frac{2}{T}\sum_{t=1}^{T}\mathbb{E}\|\theta_{q,n,0}-\theta_q\|^2$$

$$= \frac{2}{T}\sum_{t=1}^{T}\mathbb{E}\|\sum_{j=0}^{t-2}-\eta_l\tilde{g}_{q,n,j}\odot m_{q,n}\|^2+\frac{2}{T}\sum_{t=1}^{T}\mathbb{E}\|\theta_q\odot m_{n,q}-\theta_q\|^2$$

$$\leq \frac{2\eta_l^2}{T}\sum_{t=1}^{T}\mathbb{E}\|\sum_{j=0}^{t-2}(\nabla f_n(\tilde{\theta}_{q,n,j},\xi_{n,j})-\nabla f_n(\tilde{\theta}_{q,n,j})+\nabla f_n(\tilde{\theta}_{q,n,j}))\odot m_{q,n}\|^2+\frac{2}{T}\sum_{t=1}^{T}l^2\mathbb{E}\|\theta_q\|^2$$

$$\leq \frac{4\eta_l^2}{T}\sum_{t=1}^{T}\mathbb{E}\|\sum_{j=0}^{t-2}(\nabla f_n(\tilde{\theta}_{q,n,j},\xi_{n,j})-\nabla f_n(\tilde{\theta}_{q,n,j}))\odot m_{q,n}\|^2$$

$$+\frac{4\eta_l^2}{T}\sum_{t=1}^{T}\mathbb{E}\|\sum_{j=0}^{t-2}\nabla f_n(\tilde{\theta}_{q,n,j})\odot m_{q,n}\|^2+\frac{2}{T}\sum_{t=1}^{T}l^2\mathbb{E}\|\theta_q\|^2$$

$$\leq \frac{4\eta_l^2}{T}\sum_{t=1}^{T}(t-1)L^2\delta^2\sigma_l^2+\frac{2}{T}\sum_{t=1}^{T}l^2\mathbb{E}\|\theta_q\|^2+\frac{4\eta_l^2}{T}\sum_{t=1}^{T}\mathbb{E}\|\sum_{j=0}^{t-2}(\nabla f_n(\tilde{\theta}_{q,n,j})-\nabla f_n(\tilde{\theta}_q)+\nabla f_n(\tilde{\theta}_q))\odot m_{q,n}\|^2$$

$$\leq 2\eta_l^2 TL^2\delta^2\sigma_l^2+\frac{2}{T}\sum_{t=1}^{T}l^2\mathbb{E}\|\theta_q\|^2+\frac{8\eta_l^2}{T}\sum_{t=1}^{T}(t-1)\sum_{j=0}^{t-2}\mathbb{E}\|(\nabla f_n(\tilde{\theta}_{q,n,j})-\nabla f_n(\tilde{\theta}_q))\odot m_{q,n}\|^2$$

$$+\frac{8\eta_l^2}{T}\sum_{t=1}^{T}(t-1)\sum_{j=0}^{t-2}\mathbb{E}\|\nabla f_n(\tilde{\theta}_q)\odot m_{q,n}\|^2$$

$$\leq 2\eta_l^2 TL^2\delta^2\sigma_l^2+\frac{2}{T}\sum_{t=1}^{T}l^2\mathbb{E}\|\theta_q\|^2+\frac{8\eta_l^2 L^2}{T}\sum_{t=1}^{T}(t-1)\sum_{j=0}^{t-2}\mathbb{E}\|\tilde{\theta}_{q,n,j}-\tilde{\theta}_q\|^2$$

$$+8\eta_l^2 T^2\mathbb{E}\|(\nabla f_n(\tilde{\theta}_q)-\nabla f(\tilde{\theta}_q)+\nabla f(\tilde{\theta}_q))\odot m_{q,n}\|^2$$

$$\leq 2\eta_l^2 TL^2\delta^2\sigma_l^2+\frac{2}{T}\sum_{t=1}^{T}l^2\mathbb{E}\|\theta_q\|^2+\frac{16\eta_l^2 L^2}{T}\sum_{t=1}^{T}(t-1)\sum_{j=0}^{t-2}\mathbb{E}\|\theta_{q,n,j}-\theta_q\|^2+\frac{16\eta_l^2 L^2}{T}\sum_{t=1}^{T}(t-1)\sum_{j=0}^{t-2}\mathbb{E}\|\epsilon_{q,n,j}-\epsilon_q\|^2$$

$$+16\eta_l^2 T^2\mathbb{E}\|(\nabla f_n(\tilde{\theta}_q)-\nabla f(\tilde{\theta}_q))\odot m_{q,n}\|^2+16\eta_l^2 T^2\mathbb{E}\|\nabla f(\tilde{\theta}_q)\odot m_{q,n}\|^2$$

$$\leq 2\eta_l^2 TL^2\delta^2\sigma_l^2+2l^2\mathbb{E}\|\theta_q\|^2+16\eta_l^2 L^2 T^2\frac{1}{T}\sum_{t=1}^{T}\mathbb{E}\|\theta_{q,n,t-1}-\theta_q\|^2+16\eta_l^2 L^2 T^2\mathcal{E}_\epsilon+16\eta_l^2 T^2\mathcal{E}_g+16\eta_l^2 T^2\mathbb{E}\|\nabla f(\tilde{\theta}_q)\odot m_{q,n}\|^2$$

Let learning rate satisfies $16\eta_l^2 L^2 T^2\leq 1\Rightarrow\eta_l\leq\frac{1}{4LT}$, we can get $U_7$:

$$\frac{1}{T}\sum_{t=1}^{T}\mathbb{E}\|\theta_{q,n,t-1}-\theta_q\|^2\leq 2\eta_l^2 TL^2\delta^2\sigma_l^2+16\eta_l^2 L^2 T^2\mathcal{E}_\epsilon+16\eta_l^2 T^2\mathcal{E}_g+2l^2\mathbb{E}\|\theta_q\|^2+16\eta_l^2 T^2\sum_{i\in\mathcal{K}_q}\mathbb{E}\|\nabla f^i(\tilde{\theta}_q)\|^2$$

Plugging $U_7$ in $U_5$, we can get:

$$\eta_g\eta_l T\sum_{i\in\mathcal{K}_q}\mathbb{E}\|\frac{1}{T\mathcal{C}_q^i}\sum_{n\in N_q^i}\sum_{t=1}^{T}[\nabla f_n^i(\tilde{\theta}_{q,n,t-1})-\nabla f_n^i(\tilde{\theta}_q)]\|\|^2$$

$$\leq 2\eta_g\eta_l T\frac{1}{\mathcal{C}^*}\sum_{n=1}^{N}L^2[2\eta_l^2 TL^2\delta^2\sigma_l^2+16\eta_l^2 L^2 T^2\mathcal{E}_\epsilon+16\eta_l^2 T^2(3\sigma_g^2+6L^2\delta^2)$$

$$+ 2l^2\mathbb{E}\|\theta_q\|^2 + 16\eta_l^2 T^2 \sum_{i \in \mathcal{K}_q} \mathbb{E}\|\nabla f(\tilde{\theta}_q)\|^2] + 2\eta_g\eta_l T \frac{1}{\mathcal{C}^*} \sum_{n=1}^{N} L^2 \frac{1}{T} \sum_{t=1}^{T} \mathbb{E}\|\epsilon_{q,n,t-1} - \epsilon_q]\|^2$$

$$\leq 2\eta_g\eta_l T L^2 \frac{N}{\mathcal{C}^*}[2\eta_l^2 T L^2 \delta^2 \sigma_l^2 + 16\eta_l^2 L^2 T^2 \mathcal{E}_\epsilon + 16\eta_l^2 T^2 \mathcal{E}_g + 2l^2\mathbb{E}\|\theta_q\|^2 + 16\eta_l^2 T^2 \sum_{i \in \mathcal{K}_q} \mathbb{E}\|\nabla f^i(\tilde{\theta}_q)\|^2]$$

To bound $U_6$:

$$\eta_g\eta_l T \sum_{i \in \mathcal{K}_q} \mathbb{E}\|\frac{1}{T\mathcal{C}_q^i} \sum_{n \in N_q^i} \sum_{t=1}^{T} [\nabla f_n^i(\tilde{\theta}_q) - \nabla f^i(\tilde{\theta}_q)]\|^2$$

$$\leq \eta_g\eta_l T \frac{1}{T\mathcal{C}_q^i} \sum_{n \in N_q^i} \sum_{t=1}^{T} \sum_{i \in \mathcal{K}_q} \mathbb{E}\|[\nabla f_n^i(\tilde{\theta}_q) - \nabla f^i(\tilde{\theta}_q)]\|^2$$

$$\leq \eta_g\eta_l T \frac{1}{T\mathcal{C}^*} \sum_{n \in N_q^i} \sum_{t=1}^{T} \mathbb{E}\|[\nabla f_n^i(\tilde{\theta}_q) - \nabla f^i(\tilde{\theta}_q)]\|^2$$

$$\leq \frac{\eta_g\eta_l T N}{\mathcal{C}^*} \mathcal{E}_g$$

Plugging $U_5, U_6$ in $U_4$:

$$\sum_{i \in \mathcal{K}_q} \mathbb{E}[\langle \nabla f^i(\tilde{\theta}_q), -\frac{\eta_g\eta_l}{\mathcal{C}_q^i} \sum_{n \in N_q^i} \sum_{t=1}^{T} [\nabla f_n^i(\tilde{\theta}_{q,n,t-1}) - \nabla f^i(\tilde{\theta}_q)]\rangle]$$

$$\leq 2\eta_g\eta_l T L^2 \frac{N}{\mathcal{C}^*}[2\eta_l^2 T L^2 \delta^2 \sigma_l^2 + 16\eta_l^2 L^2 T^2 \mathcal{E}_\epsilon + 16\eta_l^2 T^2 \mathcal{E}_g + 2l^2\mathbb{E}\|\theta_q\|^2 + 16\eta_l^2 T^2 \sum_{i \in \mathcal{K}_q} \mathbb{E}\|\nabla f^i(\tilde{\theta}_q)\|^2]$$

$$+ \frac{\eta_g\eta_l T}{2} \sum_{i \in \mathcal{K}_q} \mathbb{E}\|\nabla f^i(\tilde{\theta}_q)\|^2 + \frac{\eta_g\eta_l T N}{\mathcal{C}^*} \mathcal{E}_g$$

Plugging $U_3, U_4$ in $U_1$:

$$\mathbb{E}[\langle \nabla f(\tilde{\theta}_q), \tilde{\theta}_{q+1} - \tilde{\theta}_q \rangle]$$

$$\leq 2\eta_g\eta_l T L^2 \frac{N}{\mathcal{C}^*}[2\eta_l^2 T L^2 \delta^2 \sigma_l^2 + 16\eta_l^2 L^2 T^2 \mathcal{E}_\epsilon + 16\eta_l^2 T^2 \mathcal{E}_g + 2l^2\mathbb{E}\|\theta_q\|^2 + 16\eta_l^2 T^2 \sum_{i \in \mathcal{K}_q} \mathbb{E}\|\nabla f^i(\tilde{\theta}_q)\|^2]$$

$$\textcolor{red}{- \frac{\eta_g\eta_l T}{4}} \sum_{i \in \mathcal{K}_q} \mathbb{E}\|\nabla f^i(\tilde{\theta}_q)\|^2 + \frac{\eta_g\eta_l T N}{\mathcal{C}^*} \mathcal{E}_g + \textcolor{red}{\frac{1}{\eta_l\eta_g T}} \delta^2$$

To bound $U_2$:

$$\frac{L}{2}\mathbb{E}\|\tilde{\theta}_{q+1} - \tilde{\theta}_q\|^2$$

$$= \frac{L}{2} \sum_{i \in \mathcal{K}_q} \mathbb{E}\|\tilde{\theta}_{q+1} - \tilde{\theta}_q\|^2 + \frac{L}{2} \sum_{i \in \mathcal{S} - \mathcal{K}_q} \mathbb{E}\|\tilde{\theta}_{q+1} - \tilde{\theta}_q\|^2$$

$$= \frac{L}{2} \sum_{i \in \mathcal{K}_q} \mathbb{E}\|\tilde{\theta}_{q+1} - \tilde{\theta}_q\|^2$$

$$= L \sum_{i \in \mathcal{K}_q} \mathbb{E}\|\theta_{q+1} - \theta_q\|^2 + L \sum_{i \in \mathcal{K}_q} \mathbb{E}\|\epsilon_{q+1} - \epsilon_q\|^2$$

$$\leq L\eta_g^2 \sum_{i \in \mathcal{K}_q} \mathbb{E}\|\Delta_q\|^2 + L\delta^2$$

$$= L\eta_g^2 \sum_{i \in \mathcal{K}_q} \mathbb{E}\| - \frac{1}{\mathcal{C}_q^i} \sum_{n \in N_q^i} \sum_{t=1}^{T} \eta_l \nabla f_n^i(\tilde{\theta}_{q,n,t-1}, \xi_{n,t-1})\|^2 + L\delta^2$$

$$\leq 3L\eta_g^2 \sum_{i \in \mathcal{K}_q} \mathbb{E}\| - \frac{1}{\mathcal{C}_q^i} \sum_{n \in N_q^i} \sum_{t=1}^{T} \eta_l [\nabla f_n^i(\tilde{\theta}_{q,n,t-1}, \xi_{n,t-1}) - \nabla f_n^i(\tilde{\theta}_{q,n,t-1})]\|^2$$

$$+ 3L\eta_g^2 \sum_{i \in \mathcal{K}_q} \mathbb{E}\| - \frac{1}{\mathcal{C}_q^i} \sum_{n \in N_q^i} \sum_{t=1}^T \eta_l [\nabla f_n^i(\tilde{\theta}_{q,n,t-1}) - \nabla f^i(\tilde{\theta}_q)]\|^2$$

$$+ 3L\eta_g^2 \sum_{i \in \mathcal{K}_q} \mathbb{E}\| - \frac{1}{\mathcal{C}_q^i} \sum_{n \in N_q^i} \sum_{t=1}^T \eta_l \nabla f^i(\tilde{\theta}_q)\|^2 + L\delta^2$$

$$\leq 3L\eta_g^2 \eta_l^2 \frac{NT}{\mathcal{C}^*} L^2 \delta^2 \sigma_l^2 + 6L\eta_g^2 \sum_{i \in \mathcal{K}_q} \mathbb{E}\| - \frac{1}{\mathcal{C}_q^i} \sum_{n \in N_q^i} \sum_{t=1}^T \eta_l [\nabla f_n^i(\tilde{\theta}_{q,n,t-1}) - \nabla f_n^i(\tilde{\theta}_q)]\|^2$$

$$+ 6L\eta_g^2 \sum_{i \in \mathcal{K}_q} \mathbb{E}\| - \frac{1}{\mathcal{C}_q^i} \sum_{n \in N_q^i} \sum_{t=1}^T \eta_l [\nabla f_n^i(\tilde{\theta}_q) - \nabla f^i(\tilde{\theta}_q)]\|^2 + 3L\eta_g^2 \sum_{i \in \mathcal{K}_q} \mathbb{E}\| - \frac{1}{\mathcal{C}_q^i} \sum_{n \in N_q^i} \sum_{t=1}^T \eta_l \nabla f^i(\tilde{\theta}_q)\|^2 + L\delta^2$$

$$\leq 3L\eta_g^2 \eta_l^2 \frac{NT}{\mathcal{C}^*} L^2 \delta^2 \sigma_l^2 + 6L\eta_g^2 \eta_l^2 \frac{NT^2}{\mathcal{C}^*} L^2 (2\eta_l^2 T L^2 \delta^2 \sigma_l^2 + 16\eta_l^2 L^2 T^2 \mathcal{E}_\epsilon + 16\eta_l^2 T^2 \mathcal{E}_g$$

$$+ 2l^2 \mathbb{E}\|\theta_q\|^2 + 16\eta_l^2 T^2 \sum_{i \in \mathcal{K}_q} \mathbb{E}\|\nabla f^i(\tilde{\theta}_q)\|^2) + 6L\eta_g^2 \eta_l^2 \frac{NT^2}{\mathcal{C}^*} \mathcal{E}_g + 3L\eta_g^2 \eta_l^2 T^2 \sum_{i \in \mathcal{K}_q} \mathbb{E}\|\nabla f^i(\tilde{\theta}_q)\|^2 + L\delta^2$$

Last we have:

$$\mathbb{E}[f(\theta_{q+1})] = \mathbb{E}[f(\tilde{\theta}_{q+1})] \leq f(\tilde{\theta}_q) + 2\eta_g \eta_l T L^2 \frac{N}{\mathcal{C}^*} [2\eta_l^2 T L^2 \delta^2 \sigma_l^2 + 16\eta_l^2 L^2 T^2 \mathcal{E}_\epsilon + 16\eta_l^2 T^2 \mathcal{E}_g$$

$$+ 2l^2 \mathbb{E}\|\theta_q\|^2 + 16\eta_l^2 T^2 \sum_{i \in \mathcal{K}_q} \mathbb{E}\|\nabla f^i(\tilde{\theta}_q)\|^2] \textcolor{red}{- \frac{\eta_g \eta_l T}{4} \sum_{i \in \mathcal{K}_q} \mathbb{E}\|\nabla f^i(\tilde{\theta}_q)\|^2} + \frac{\eta_g \eta_l T N}{\mathcal{C}^*} \mathcal{E}_g$$

$$+ 3L\eta_g^2 \eta_l^2 \frac{NT}{\mathcal{C}^*} L^2 \delta^2 \sigma_l^2 + 6L^3 \eta_g^2 \eta_l^2 \frac{NT^2}{\mathcal{C}^*} (2\eta_l^2 T L^2 \delta^2 \sigma_l^2 + 16\eta_l^2 L^2 T^2 \mathcal{E}_\epsilon + 16\eta_l^2 T^2 \mathcal{E}_g + 2l^2 \mathbb{E}\|\theta_q\|^2$$

$$+ 16\eta_l^2 T^2 \sum_{i \in \mathcal{K}_q} \mathbb{E}\|\nabla f^i(\tilde{\theta}_q)\|^2) + 6L\eta_g^2 \eta_l^2 \frac{NT^2}{\mathcal{C}^*} \mathcal{E}_g + 3L\eta_g^2 \eta_l^2 T^2 \sum_{i \in \mathcal{K}_q} \mathbb{E}\|\nabla f^i(\tilde{\theta}_q)\|^2 + L\delta^2 + \frac{1}{\eta_g \eta_l T} \delta^2$$

$$= f(\tilde{\theta}_q) + \eta_g \eta_l T[-\frac{1}{4} + 3L\eta_g \eta_l T + 16\eta_l^2 T^2 (2L^2 \frac{N}{\mathcal{C}^*} + 6L^3 \eta_g \eta_l \frac{NT}{\mathcal{C}^*})] \sum_{i \in \mathcal{K}_q} \mathbb{E}\|\nabla f^i(\tilde{\theta}_q)\|^2$$

$$+ \eta_g \eta_l T[2l^2 (2L^2 \frac{N}{\mathcal{C}^*} + 6L^3 \eta_g \eta_l \frac{NT}{\mathcal{C}^*})] \mathbb{E}\|\theta_q\|^2$$

$$+ \eta_g \eta_l T (2L^2 \frac{N}{\mathcal{C}^*} + 6L^3 \eta_g \eta_l \frac{NT}{\mathcal{C}^*})(2\eta_l^2 T L^2 \delta^2 \sigma_l^2 + 16\eta_l^2 L^2 T^2 \mathcal{E}_\epsilon + 16\eta_l^2 T^2 \mathcal{E}_g)$$

$$+ \eta_g \eta_l T (\frac{N}{\mathcal{C}^*} + 6L\eta_g \eta_l \frac{NT}{\mathcal{C}^*}) \mathcal{E}_g + 3L\eta_g^2 \eta_l^2 \frac{NT}{\mathcal{C}^*} L^2 \delta^2 \sigma_l^2 + L\delta^2 + \frac{1}{\eta_g \eta_l T} \delta^2$$

$$\overset{a}{\leq} f(\tilde{\theta}_q) \textcolor{red}{- \frac{\eta_g \eta_l T}{16} \sum_{i \in \mathcal{K}_q} \mathbb{E}\|\nabla f^i(\tilde{\theta}_q)\|^2} + \Phi$$

where $a$ follows because:

$$32\eta_l^2 T^2 \frac{N}{\mathcal{C}^*} L^2 \leq \frac{1}{16} \Rightarrow \eta_l \leq \frac{\sqrt{\mathcal{C}^*}}{16TL\sqrt{N}}$$

$$\textcolor{red}{96L^3 \eta_l^3 \eta_g T^3 \frac{N}{\mathcal{C}^*} \leq \frac{1}{16} \Rightarrow \eta_g \leq \frac{2\sqrt{N}}{\sqrt{\mathcal{C}^*}}}$$

$$\textcolor{red}{3L\eta_l \eta_g T \leq \frac{1}{16} \Rightarrow \eta_l \eta_g \leq \frac{1}{48TL}.}$$

Thus we can get the following inequality.

$$\frac{\eta_g \eta_l T}{16} \sum_{q=1}^Q \sum_{i \in \mathcal{K}_q} \mathbb{E}\|\nabla f^i(\tilde{\theta}_q)\|^2 \leq \mathbb{E}[f(\theta_1)] + 4\eta_g \eta_l T l^2 \frac{L^2 N}{\mathcal{C}^*} \sum_{i=1}^Q \mathbb{E}\|\theta_q\|^2 + QL\delta^2 + \frac{Q}{\eta_g \eta_l T} \delta^2$$

$$+ \frac{\eta_g \eta_l T L^2 Q}{8} (\frac{1}{8T} \delta^2 \sigma_l^2 + \mathcal{E}_\epsilon + \frac{1}{L^2} \mathcal{E}_g) + 2\eta_g \eta_l T Q (\frac{N}{\mathcal{C}^*}) \mathcal{E}_g + \eta_g \eta_l T Q \frac{L^2 N}{16T\mathcal{C}^*} \delta^2 \sigma_l^2$$

Dividing both sides above by $\frac{T\eta_g\eta_l Q}{16}$ we can get

$$\frac{1}{Q}\sum_{q=1}^{Q}\sum_{i\in\mathcal{K}_q}\mathbb{E}\|\nabla f^i(\tilde{\theta}_q)\|^2 \leq \frac{16\mathbb{E}[f(\theta_1)]}{T\eta_l\eta_g Q} + 64l^2(\frac{L^2N}{\mathcal{C}^*})\frac{1}{Q}\sum_{i=1}^{Q}\mathbb{E}\|\theta_q\|^2 + 2L^2(\frac{1}{8T}\delta^2\sigma_l^2 + \mathcal{E}_\epsilon + \frac{1}{L^2}\mathcal{E}_g)$$

$$+ \frac{32N}{\mathcal{C}^*}\mathcal{E}_g + \frac{L^2N}{T\mathcal{C}^*}\delta^2\sigma_l^2 + \frac{16L}{T\eta_l\eta_g}\delta^2 + \frac{16}{\eta_g^2\eta_l^2 T^2}\delta^2$$

Supposing that the step size $\eta_l = \frac{1}{\sqrt{Q}}, \eta_g = \frac{\sqrt{\mathcal{C}^*}}{\sqrt{T}}$, when the constant $C > 0$ exists, and perturbation amplitude $\delta$ proportional to the learning rate, e.g., $\delta = \frac{1}{\sqrt{Q}}$ ,the convergence rate can be expressed as follows:

$$\frac{1}{Q}\sum_{q=1}^{Q}\sum_{i\in\mathcal{K}_q}\mathbb{E}\|\nabla f^i(\theta_q)\|^2 \leq \mathcal{O}(\frac{A_0}{\sqrt{QT\mathcal{C}^*}} + \frac{l^2 B_0}{\mathcal{C}^*} + \frac{\sigma_g^2}{\mathcal{C}^*} + \frac{\sigma_l^2}{TQ} + \frac{\sigma_l^2}{TQ\mathcal{C}^*} + \frac{1}{Q\mathcal{C}^*} + \frac{1}{\mathcal{C}^* T} + \frac{1}{\sqrt{TQ\mathcal{C}^*}} + \frac{1}{Q})$$

where $A_0 = \mathbb{E}[f(\theta_1)], B_0 = \frac{1}{Q}\sum_{i=1}^{Q}\mathbb{E}[f(\theta_q)]$.

## D  GENERALIZATION BOUND

***Proof of Lemma 5***  Let $\Delta_i = \left|f^i(\theta\odot m + \epsilon, X) - f^i(\theta, X)\right|_2$. We will prove using induction that for any $i \geq 0$:

$$\Delta_i \leq \prod_{j=1}^{i}\left(s_j + \frac{1}{r}\right)\left(\prod_{j=1}^{i}\|\theta_j\|_2\right)|X|_2\sum_{j=1}^{i}\frac{\|\epsilon_j\|_2}{\|\theta_j\|_2}.$$

The induction base holds clearly since $\Delta_0 = |X - X|_2 = 0$. For any $i \geq 1$, we have the following:

$$\Delta_{i+1} = \left|(\theta_{i+1}\odot m_{i+1} + \epsilon_{i+1})\phi_i(f^i(\theta_i\odot m_i + \epsilon_i, X)) - \theta_{i+1}\phi_i(f^i(\theta, X))\right|_2$$

$$= \left|(\theta_{i+1}\odot m_{i+1} + \epsilon_{i+1})\left(\phi_i(f^i(\theta_i\odot m_i + \epsilon_i, X)) - \phi_i(f^i(\theta, X))\right) + \epsilon_{i+1}\phi_i(f^i(\theta, X))\right|_2$$

$$\leq \left(\|\theta_{i+1}\odot m_{i+1}\|_2 + \|\epsilon_{i+1}\|_2\right)\left|\phi_i(f^i(\theta_i\odot m_i + \epsilon_i, X)) - \phi_i(f^i(\theta, X))\right|_2 + \|\epsilon_{i+1}\|_2\left|f^i(\theta, X))\right|_2$$

$$\overset{a}{\leq} \left(\|\theta_{i+1}\odot m_{i+1}\|_2 + \|\epsilon_{i+1}\|_2\right)\left|f^i(\theta_i\odot m_i + \epsilon_i, X) - f^i(\theta, X)\right|_2 + \|\epsilon_{i+1}\|_2\left|f^i(\theta, X)\right|_2$$

$$= \Delta_i\left(\|\theta_{i+1}\odot m_{i+1}\|_2 + \|\epsilon_{i+1}\|_2\right) + \|\epsilon_{i+1}\|_2\left|f^i(\theta, X)\right|_2,$$

where $a$ follows Lipschitz property of the activation function and using $\phi(0) = 0$. The $\ell_2$ norm of outputs of layer $i$ is bounded by $|X|_2\prod_{j=1}^{i}\|\theta_j\|_2$ and by the lemma assumption we have $\|\epsilon_{i+1}\|_2 \leq \frac{1}{r}\|\theta_{i+1}\|_2$. Let $s_j$ be the remaining rate of $j$-th layer, $\|\theta_j\odot m_j\|_2 = s_j\|\theta_j\|$. Therefore, using the induction step, we get the following bound:

$$\Delta_{i+1} \leq \Delta_i\left(s_{i+1} + \frac{1}{r}\right)\|\theta_{i+1}\|_2 + \|\epsilon_{i+1}\|_2|X|_2\prod_{j=1}^{i}\|\theta_j\|_2$$

$$\leq \prod_{j=1}^{i+1}\left(s_j + \frac{1}{r}\right)\left(\prod_{j=1}^{i+1}\|\theta_j\|_2\right)|X|_2\sum_{j=1}^{i}\frac{\|\epsilon_j\|_2}{\|\theta_j\|_2} + \frac{\|\epsilon_{i+1}\|_2}{\|\theta_{i+1}\|_2}|X|_2\prod_{j=1}^{i+1}\|\theta_i\|_2$$

$$\leq \prod_{j=1}^{i+1}\left(s_j + \frac{1}{r}\right)\left(\prod_{j=1}^{i+1}\|\theta_j\|_2\right)|X|_2\sum_{j=1}^{i+1}\frac{\|\epsilon_j\|_2}{\|\theta_j\|_2}.$$

Let the norm of input $X$ be bounded by $A, A > 0$, we can gain

$$|f(\theta\odot m + \epsilon) - f(\theta)|_2 \leq A\prod_{j=1}^{r}(s_j + \frac{1}{r})\prod_{j=1}^{r}\|\theta_j\|_2\sum_{j=1}^{r}\frac{\|\epsilon_j\|_2}{\|\theta_j\|_2}.$$

**Proof of Theorem 2**  Since $\epsilon \sim \mathcal{N}(0, \sigma^2 I)$, we get the following bound for the spectral norm of $\epsilon_i$:

$$\mathbb{P}_{\epsilon_i \sim N(0,\sigma^2 I)}\left[\|\epsilon_i\|_2 > t\right] \leq 2\tilde{h}e^{-t^2/2\tilde{h}\sigma^2}.$$

Taking a union bond over the layers, we get that, with probability $\geq \frac{1}{2}$, the spectral norm of the perturbation $\epsilon_i$ in each layer is bounded by $\sigma\sqrt{2\tilde{h}\ln(4r\tilde{h})}$. Let $\beta = (\prod_{j=1}^{r}\|\theta_j\|_2)^{1/r}$, denoting the geometric mean of the $\theta$'s spectral norm across all layers. Plugging this spectral norm bound into Lemma 5 we have that with probability at least $\frac{1}{2}$,

$$\max_X |f(\theta \odot m + \epsilon, X) - f(\theta, X)|_2 \leq A\prod_{j=1}^{r}(s_j + \frac{1}{r})\beta^r \sum_i \frac{\|\epsilon_i\|_2}{\beta}$$

$$= A\prod_{j=1}^{r}(s_j + \frac{1}{r})\beta^{r-1} \sum_i \|\epsilon_i\|_2$$

$$\leq erA\prod_{j=1}^{r}(s_j + \frac{1}{r})^{r-1}\sigma\sqrt{2\tilde{h}\ln(4r\tilde{h})} \leq \frac{\gamma}{4},$$

where we choose $\sigma = \frac{\gamma}{16\prod_{j=1}^{r}(s_j + \frac{1}{r})rA\tilde{\beta}^{r-1}\sqrt{\tilde{h}\ln(4\tilde{h}r)}}$ to get the last inequality. Hence, the perturbation $\epsilon$ with the above value of $\sigma$ satisfies the assumptions of the Lemma 4.

We now calculate the KL-term in Lemma 4 with the chosen distributions for $P$ and $\epsilon$, for the above value of $\sigma$.

$$KL(\theta + \epsilon \| P) \leq \frac{|\theta|^2}{2\sigma^2} = \frac{16^2 \prod_{j=1}^{r}(s_j + \frac{1}{r})^2 r^2 A^2 \tilde{\beta}^{2r-2}\tilde{h}\ln(4\tilde{h}r)}{2\gamma^2} \sum_{i=1}^{r} \|\theta_i\|_F^2$$

$$\leq \mathcal{O}\left(\prod_{j=1}^{r}(s_j + \frac{1}{r})^2 A^2 r^2 \tilde{h}\ln(\tilde{h}r)\frac{\beta^{2r}}{\gamma^2}\sum_{i=1}^{r}\frac{\|\theta_i\|_F^2}{\beta^2}\right)$$

$$\leq \mathcal{O}\left(\prod_{j=1}^{r}(s_j + \frac{1}{r})^2 A^2 r^2 \tilde{h}\ln(\tilde{h}r)\frac{\prod_{i=1}^{r}\|\theta_i\|_2^2}{\gamma^2}\sum_{i=1}^{r}\frac{\|\theta_i\|_F^2}{\|\theta_i\|_2^2}\right).$$

Hence, for any $\tilde{\beta}$, with probability $1 - \zeta$ and for all $\theta$ such that, $|\beta - \tilde{\beta}| \leq \frac{1}{r}\beta$, we have:

$$\mathcal{L}(f(\theta)) \leq \hat{\mathcal{L}}_\gamma(f(\theta)) + \mathcal{O}\left(\sqrt{\frac{\prod_{j=1}^{r}(s_j + \frac{1}{r})^2 A^2 r^2 \tilde{h}\ln(\tilde{h}r)\prod_{i=1}^{r}\|\theta_i\|_2^2 \sum_{i=1}^{r}\frac{\|\theta_i\|_F^2}{\|\theta_i\|_2^2} + \ln\frac{d}{\zeta}}{d\gamma^2}}\right). \quad (22)$$

Considering the assumption in Theorem 2, any layer $\theta_j$ satisfies $\frac{1}{M} \leq \frac{\|\theta_j\|_2}{\beta} \leq M$, and approximation $\tilde{\beta}$ satisfies $\left|\beta - \tilde{\beta}\right| \leq \frac{1}{r}\beta$. Thus, we use a cover of size $\mathcal{O}((rlogM)^r)$. For $\zeta > 0$ with probability $1 - \zeta$, we have:

$$\mathcal{L}(f(\theta)) \leq \hat{\mathcal{L}}_\gamma(f(\theta)) + \mathcal{O}(\sqrt{\frac{\prod_{j=1}^{r}(s_j + \frac{1}{r})^2 r^2 A^2 \ln(r\tilde{h})\prod_{j=1}^{r}\|\theta_j\|_2^2 \sum_{i=j}^{r}\frac{s_j\|\theta_j\|_F^2}{\|\theta_j\|_2^2} + r\ln\frac{rd\log M}{\zeta}}{d\gamma^2}}),$$

$$(23)$$

In order to apply the above result in the distributed scenario with N clients, we apply a union bound to have the bound hold simultaneously for the distribution of each client. For $\zeta > 0$ with probability $1 - \zeta$, we have:

$$\mathcal{L}(f(\theta)) \leq \hat{\mathcal{L}}_\gamma(f(\theta)) + \mathcal{O}(\sqrt{\frac{\prod_{j=1}^{r}(s_j + \frac{1}{r})^2 r^2 A^2 \ln(r\tilde{h})\prod_{j=1}^{r}\|\theta_j\|_2^2 \sum_{i=j}^{r}\frac{s_j\|\theta_j\|_F^2}{\|\theta_j\|_2^2} + r\ln\frac{Nrd\log M}{\zeta}}{d\gamma^2}}).$$

$$(24)$$

## E    ADDITIONAL EXPERIMENTAL DETAILS

**Backbone.** We use two ViT (Dosovitskiy et al., 2020) variants with different capabilities, as shown in Tab.4.

Table 4: Details of used Vision Transformer model.

| Model | Layer | Hidden size | MLP size | Heads | Params | FLOPs |
|-------|-------|-------------|----------|-------|--------|-------|
| ViT-Small | 4 | 64 | 128 | 8 | 0.64MB | 2.637M |
| ViT | 12 | 192 | 768 | 8 | 21.34MB | 92.781M |

**Impact of model architecture.** In order to show the effectiveness of proposed method, we additionally conduct experiments using different architecture model ResNet18 on CIFAR-10 with Dirichlet distribution ($\mu = 1.0$). The results are shown in Table 5 and our proposed method is the best.

Table 5: Comparison of methods on different model architectures.

| | Methods | ResNet18/CIFAR-10 | ViT-Small/CIFAR-10 | ViT/CIFAR-100 |
|---|---------|-------------------|--------------------|---------------|
| | Centralized | 81.02 | 59.19 | 35.61 |
| Full | FedAvg | 76.78(2.92) | 55.70(2.96) | 30.95(2.90) |
| | FedSAM | 78.35(2.89) | 57.03(2.62) | 32.11(2.68) |
| | IST | 62.51(5.39) | 38.06(10.27) | 17.15(5.54) |
| | OAP | 64.90(3.54) | 48.29(9.29) | 26.09(6.97) |
| Sub. | PruneFL | 64.04(4.06) | 48.20(5.16) | 22.35(5.80) |
| | FedRolex | 65.30(3.16) | 44.84(4.75) | 21.73(2.57) |
| | RAM-Fed | 69.15(3.15) | 50.19(4.16) | 23.25(4.77) |
| | SubDisMO | 76.34(3.33) | 51.23(4.77) | 25.43(4.56) |

**Impact of mask policy.** In our method design, we give the random mask as a mask policy example and present the theoretical analysis. It shows the superior convergence rate and generalization error bound of the proposed method in solving the general distributed minimax optimization problem. And based on SubDisMO, we can change to any submodel construction method to improve the empirical performance. Different mask policy would lead to different $\mathcal{C}^*$ so the model convergence rate is different, even different empirical performance. Here, we compare with rolling mask policy with different step and the overlap rate. When the new local submodel overlaps 50% from the last one, the performance is better than no overlap. It's suggested based on our proposed SubDisMO choose appropriate mask policy for practical application.

Table 6: Experimental results on different mask policies.

| Mask policy | CIFAR-10($\mu = 1.0$) | CIFAR-10(IID) |
|-------------|------------------------|----------------|
| Random | 51.23(4.77) | 55.99(1.85) |
| Rolling-no overlap | 45.11(5.79) | 47.35(1.70) |
| Rolling-50% overlap | 51.57(4.17) | 55.48(1.36) |

**Computational efficiency.** Considering that each deep network includes multiple operations and computations, we commonly use the amount of computation (FLOPS) to analyze the time complexity and the number of parameters to analyze the space complexity. The results of each process are shown in Table 7. In comparison to state-of-the-art distributed learning algorithm, FedSAM, where each client needs to train the full model, our proposed method allows each client to train only a submodel. This significantly reduces both the computational load and the number of parameters, thereby improving efficiency in terms of both computation and storage.

**Scalability.** In order to explore the scalability of our proposed algorithm *SubDisMO*, we add the experiment that the number of clients is 10, 100, 1000 on *CIFAR-10*. And we repartition the data for each client under Dir($\mu = 1.0$). Considering insufficient number of data and practical large-scale distributed systems, we use data reply method for clients instead of no repeated division. For

Table 7: Computation comparison of SubDisMO and FedSAM with full model training.

| Method | Parameters | FLOPs |
|---|---|---|
| Ours(25% submodel/Vit-Small) | 55.242K | 103.809M |
| Ours(50% submodel/Vit-Small) | 88.522K | 181.24M |
| FedSAM with full Vit-Small | 155.082K | 336.102M |
| Ours(25% submodel/Vit) | 1.599M | 3.047G |
| Ours(50% submodel/Vit) | 2.93M | 5.99G |
| FedSAM with full Vit | 5.597M | 11.876G |

10 clients, each client maintains about 5,000 data samples, for 100 clients, each client maintains about 3,000 data samples, and for 1,000 clients, each client maintains about 500 data samples. The results are shown in Table 8. We can find that as the number of clients increasing, the performance decreases. The main reason is the local amount of data samples is decreased, which further affect the performance of the local model.

Table 8: Experimental results on scalability studies of SubDisMO.

| Method | 10-Clients | 100-Clients | 1000-Clients |
|---|---|---|---|
| RAM-Fed | 50.19(4.16) | 37.09(8.46) | 27.02(9.88) |
| Ours | 51.23(4.77) | 47.53(6.41) | 32.05(8.97) |

To show the effectiveness of our proposed SubDisMO, we give the convergence curve in large-scale distributed systems, as shown in Figure 5

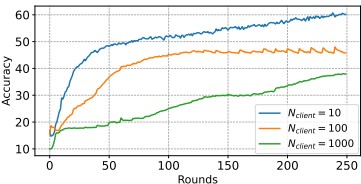

Figure 5: Training process of different number of clients.

## F    ADDITIONAL DISCUSSION

In this section, we discuss the limitations and broader impacts of the work.

**Limitations.** Although we provide rigorous theoretical proof and extensive experiments analysis, the experiments are mainly conducted in computer vision (CV) scenario. We leave the implementation of other tasks, such as natural language process (NLP) task as a future research exploration.

**Broader Impacts.** The resource-aware distributed minimax optimization algorithm, SubDisMO, significantly advances the field of machine learning by enabling efficient training in resource-limited environments through adaptive submodel pruning. This adaptability ensures scalability to large-scale models, making advanced machine learning techniques accessible for a broader range of applications and devices. This advancement can allow organizations and individuals with limited computational resources to participate in and benefit from cutting-edge AI developments. The smaller institutions and research groups can engage in large-scale model training without the need for expensive hardware investments. Overall, our work paves the way for more inclusive and sustainable AI development, fostering innovation and collaboration across various communities with otherwise limited computational resources.

