# OpenReview forum: "Generalized Resource-Aware Distributed Minimax Optimization"
_ICLR.cc/2025/Conference — Submitted to ICLR 2025_

### Official Review · Reviewer_Wn7T · 2024-10-30

**Soundness:** 3
**Presentation:** 2
**Contribution:** 3
**Rating:** 5
**Confidence:** 3

**Summary:**

This paper proposes a distributed minimax optimization algorithm, SubDisMO, assigning adaptive-sized submodels to the clients based on the resource budgets, thus enhancing the generalization of the global model by training adaptive submodels with perturbations. Additionally, the paper provides the theoretical analysis of convergence and generalization bound of SubDisMO, and experimental results demonstrate its superior effectiveness compared to state-of-the-art methods.

**Strengths:**

1. SubDisMO is the first distributed minimax optimization algorithm to account for resource budgets, enabling the training of resource-adaptive submodels with perturbations.
2. The paper provides the theoretical analysis of the convergence rate and generalization bound of SubDisMO.
3. It includes an in-depth analysis of different heterogeneity of client data, minimum covering number and the upper bound of the perturbation to evaluate the effectiveness of SubDisMO.

**Weaknesses:**

1. The formulation of SubDisMO as a distributed minimax optimization problem needs further clarification. Specifically, in Equation 2 on page 2, it is unclear whether, after altering the constraints of the minimax problem, Equation 2 still can be recognized as a minimax problem. This raises questions about whether SubDisMO functions as a distributed minimax optimization problem or more as a solution to minimax problems.
2. While SubDisMO claims that it’s a resource-aware distributed minimax optimization algorithm, it lacks analysis on the impact of varying resource budgets, particularly concerning the influence of mask distribution in SubDisMO.
3. Some figures in the paper need clearer presentations, for example, Figure 2 and Figure 4 need to be clear enough to see the difference between each method setting.

**Questions:**

1. SubDisMO designs a novel distributed minimax optimization problem, which introduces an inner maximization on perturbation $\epsilon_i$ for each client submodel. I have a question about this definition in Eq.2 on page 2: after changing the constraints of the minimax problem, can Eq.2 be recognized as a minimax problem? Is SubDisMO more like a solution for distributed minimax optimization problems?
2. In SubDisMO, the masking policy of the submodel is random, I have a question: if the masking policy is rolling, whether SubDisMO can still work in this setting? It means the performance of rolling mask with perturbation. Is the performance of this combination better than the current random mask with perturbation?

---

> ### Author Response · Authors · 2024-11-21
>
> We thank the reviewer Wn7T for the time and valuable feedback! We would try our best to address the comments one by one.
>
> **Response to Weakness1 & Question1:**
>
> Thank you for your insightful comments. We would like to clarify that both Eq. 1 and Eq. 2 represent distributed minimax optimization problems. In Eq. 1 (page 1), we present the general formulation of the distributed minimax problem, which aims to minimize the maximum average loss across all clients. Building on this, Eq. 2 (page 2) incorporates resource constraints and extends the formulation to a general global model optimization. The constraint in Eq. 2 is specifically designed to address the limited resources available at the local client level, while from a global perspective, the optimization goal remains to solve a distributed minimax problem.
> And we admit that SubDisMO is a solution for solving a distributed minimax optimization problem.
>
> **Response to Weakness2:**
>
> Thank you very much for the insightful and valuable comments.
>
> (1) In Section 3, we define that the resource-aware adaptive mask $m_{q,n}$ is generated from policy $P(\theta_q; R_n)$, where $R_n$ represents the resource constraints of client $n$.
>
> (2) From the theoretical aspect, resource budget affects the minimum covering rate $\mathcal{C}^*$ and the noise induced from the mask bounded by $l$. As shown in Corollary 1, $\frac{1}{Q}\sum_{q=1}^Q \sum_{i \in \mathcal{K} _ q} \mathbb{E}\left[\|\nabla f^i(\theta _ q)\|^2\right] \leq \mathcal{O}\left(\frac{A _ 0}{\sqrt{QT\mathcal{C}^*}} + \frac{l^2 B _ 0}{\mathcal{C}^*} + \frac{\sigma _ g^2}{\mathcal{C}^*} + \frac{\sigma _ l^2}{TQ} + \frac{1}{\sqrt{TQ\mathcal{C}^*}}\right)$, high budgets lead to more clients training larger submodels, so that $\mathcal{C}^*$ is larger, and $l$ is smaller, which will speed up the model convergence.
>
> (3) From the empirical experimental aspect, we evaluate the impact of varying resource budgets. Section 5.3 presents the impact of $\mathcal{C}^*$. And when the resource budgets across clients are high, all clients train bigger submodel, so $\mathcal{C}^*$ is larger, and vice versa. Figure 4 shows when the resource budget is high, the larger $\mathcal{C}^*$ is, the performance and convergence rate are both improved.
>
> **Response to Weakness3:**
>
> We are sorry for our unclear presentations. We improve the presentation of the experiment figures and the modified ones are presented in the revised paper.
>
> **Response to Question2:**
>
> Thanks for your detailed comments.
>
> (1) In our method design, we give the random mask as a mask policy example and present the theoretical analysis. It shows the superior convergence rate and generalization error bound of the proposed method in solving the general distributed minimax optimization problem. And based on SubDisMO, we can change to any submodel construction method.
>
> (2) Following your advice, we change the policy to a rolling mask, and the results are shown as follows. We can see that although we changed the rolling mask policy, the performance is not absolutely better. The rolling step and the overlap rate also impact the performance of the global model. When the new local submodel overlaps 50\% from the last one, the performance is better than no overlap. That's because different policies lead to different $\mathcal{C}^*$, and as we present in Corollary 1, a larger $\mathcal{C}^*$ leads to a better convergence rate, which further effects the model performance.
>
> |  **Mask policy** 	| **CIFAR-10($\mu = 1.0$)**| **CIFAR-10(IID)** |
> |:------------:|:-------:|:-------:|
> | Random| 51.23(4.77) | 55.99(1.85) |
> | Rolling-no overlap| 45.11(5.79) | 47.35(1.70) |
> | Rolling-50% overlap| 51.57(4.17) | 55.48(1.36) |
>
> If there are any further questions, we are happy to clarify and try to address them.

---

> ### Author Response · Authors · 2024-11-27
> **Window for discussion is closing**
>
> Dear Reviewer Wn7T,
>
> Thanks a lot for your time in reviewing and reading our response and the revision. Thanks very much for your valuable comments. We sincerely understand you're busy. But as the window for discussion is closing, would you mind checking our responses and confirming whether you have any further questions? We look forward to answering more questions from you.
>
> Best, Authors

---

> ### Comment · Reviewer_Wn7T · 2024-11-27
>
> Thanks for your response. The results of rolling masking and random masking are confusing to me. In [1], the rolling masking is better than the random masking. Besides, in their experiments, they did not consider the ratio of overlapping. Can authors make further explanations about this?
>
>
> [1] Alam, Samiul, et al. "Fedrolex: Model-heterogeneous federated learning with rolling sub-model extraction." Advances in neural information processing systems 35 (2022): 29677-29690.

---

> > ### Author Response · Authors · 2024-11-28
> >
> > Thank you very much for your comments. We need to explain the experimental settings further.
> >
> > **FedRolex**: we randomly split the model into 4 parts at the beginning of training, then in steps of 1/4 model to choose the next submodel.
> >
> > **Ours**: we split the model into 4 parts according to the rank of parameters in each layer at the beginning of every communication round, then each client arbitrarily selects 1 part or 2 parts to form the local submodel based on the local resource.
> >
> > Last we want to claim different policies lead to different $\mathcal{C}^*$. In training process, the $\mathcal{C}^*$ of our method may better than FedRolex, thus we can gain the better convergence rate and performance.
> >
> > We will be very happy to clarify further concerns (if any). If we have resolved all your concerns, we would greatly appreciate it if you could raise the score.

---

### Official Review · Reviewer_diba · 2024-11-01

**Soundness:** 4
**Presentation:** 3
**Contribution:** 4
**Rating:** 8
**Confidence:** 2

**Summary:**

This paper introduces SubDisMO, a resource-aware distributed minimax optimization algorithm designed to enhance the generalization capability of global models in resource-limited environments. SubDisMO mitigates model sharpness by using perturbations to adaptively train submodels, thereby improving overall model performance. The authors provide a rigorous theoretical analysis, proving that SubDisMO achieves an asymptotically optimal convergence rate of $O(1/\sqrt{QTC^*})$ under non-convex conditions, along with a tighter generalization bound based on perturbation and parameter retention rates in each layer. Extensive experiments on CIFAR-10 and CIFAR-100 demonstrate that SubDisMO outperforms existing state-of-the-art methods in resource-constrained training scenarios, validating its effectiveness and superior generalization capabilities.

**Strengths:**

Originality: SubDisMO introduces a novel approach by integrating distributed optimization, model pruning, and adversarial training, addressing the challenge of training large-scale models in resource-constrained environments.

Quality: The research is of high quality, with rigorous theoretical analysis and extensive experiments that validate the algorithm's effectiveness against state-of-the-art methods on standard datasets.

Clarity: The paper is well-organized and clearly articulates the problem, methodology, and results, with figures and tables that effectively support the presentation.

Significance: SubDisMO's ability to train large models efficiently in distributed settings with limited resources is significant for edge computing and privacy-preserving ML, potentially impacting a broad range of applications.

**Weaknesses:**

## Complexity Analysis:
The paper could further improve by providing a more detailed analysis of the computational complexity of SubDisMO compared to other methods. While the focus is on communication efficiency, understanding the trade-offs in terms of computational resources could provide a more comprehensive view of the efficiency gains.

## Scalability Concerns:
Although the paper addresses the challenge of limited resources, there is room for a deeper discussion on the scalability of SubDisMO in extremely large-scale distributed systems with potentially thousands of clients. Addressing how the algorithm performs as the number of clients grows would be valuable.

**Questions:**

## Computational Complexity:
How does the computational complexity of SubDisMO compare to other state-of-the-art distributed learning algorithms?

## Scalability in Large-Scale Systems:
What are the expected performance and convergence characteristics of SubDisMO when the number of clients increases to several thousands? Are there any bottlenecks or challenges anticipated?

---

> ### Author Response · Authors · 2024-11-21
>
> First, we sincerely thank you very much for your recognition of our work.  We would try our best to improve our work following the comments.
>
> **Response to Weakness1&Question1:**
>
> Thank you very much for the insightful comments. Considering that each deep network includes multiple operations and computations, we commonly use the amount of computation (FLOPS) to analyze the time complexity and the number of parameters to analyze the space complexity. The results of each process are shown as follows. In comparison to state-of-the-art distributed learning algorithms, where each client needs to train the full model, our proposed method allows each client to train only a submodel. This significantly reduces both the computational load and the number of parameters, thereby improving efficiency in terms of both computation and storage.
>
> |  **Methods** 	| **#Parameters** |**FLOPs** |
> |:------------:|:-------:|:-------:|
> | Ours(25\% submodel/Vit-Small) | 55.242K | 103.809M |
> | Ours(50\% submodel/Vit-Small) | 88.522K | 181.24M |
> | FedSAM with full Vit-Small  | 155.082K | 336.102M |
> | Ours(25\% submodel/Vit) | 1.599M | 3.047G |
> | Ours(50\% submodel/Vit) | 2.93M | 5.99G |
> | FedSAM with full Vit | 5.597M | 11.876G |
>
>
> **Response to Weakness2&Question2:**
>
> Thank you very much for the insightful comments. Here we explore the scalability of our proposed algorithm SubDisMO. In these experiments, we test scenarios with 100 and 1000 clients. We repartition the data for each client under Dir($\alpha=1.0$) on CIFAR-10. To compare scalability performance, we conduct the same experiments on RAM-Fed. The figure about the convergence curve is added on Page 21 in the revised paper. And the numerical results are shown as follows. It's noticed that as the number of clients increases, the performance decreases. The reason is that as the number of clients increases, the number of data in each client will decrease, which further affects the performance of the local model. In summary, it demonstrates that our proposed SubDisMO can scale well in extremely large-scale distributed systems.
>
> |  **Methods** 	| **10-Clients**| **100-Clients** |**1000-Clients** |
> |:------------:|:-------:|:-------:|:-------:|
> | RAM-Fed| 50.19(4.16) | 37.09(8.46) | 27.02(9.88) |
> | Ours| 51.23(4.77) | 47.53(6.41) | 32.05(8.97) |
>
> If there are any further questions, we are happy to clarify and try to address them.

---

> > ### Comment · Reviewer_diba · 2024-11-24
> >
> > Thank you for the detailed response, particularly the additional comparison to sota algorithms and experiments of scalability . I will maintain my previous score and would be glad to see this work accepted.

---

> > > ### Author Response · Authors · 2024-11-24
> > >
> > > We really appreciate you taking time to read and respond to our rebuttal. And thank you very much again for recognizing our work.

---

### Official Review · Reviewer_haCk · 2024-11-04

**Soundness:** 2
**Presentation:** 3
**Contribution:** 2
**Rating:** 3
**Confidence:** 4

**Summary:**

This paper studies distributed minimax optimization problem in the context of resource-limited clients. The paper introduces a generalized resource-aware distributed minimax optimization algorithm, which prunes the global large-scale model into adaptive-sized submodels with perturbations to accommodate varying resources during each communication round. The theoretically results demonstrate asymptotically optimal convergence and provide a generalization bound corresponding to the parameters. Experiments on CIFAR-10 and CIFAR-100 demonstrate the superior generalization and effectiveness of the algorithm.

**Strengths:**

This paper is well-written with clear objectives and contributions. It is the first time that a resource-limited distributed method was proposed for the minimax problem. The proposed algorithm adaptively generates submodels through local resources and trains with perturbations, which enhances the generality of the global model. The theoretical result provides a convergence bound that matches the sota result for full model settings and a generalization bound corresponding to the perturbation and parameter remaining rate in each layer.
Experiments verifies the results.

**Weaknesses:**

My major concern about this paper lies in the assumptions on the stepsize $\eta_l$ and $\eta_g$ in the theoretical analysis.
i). The conditions in Theorem 1 are somehow self-contradictory. According to line 3 - 6 in the big blanket, $\eta_g\leq\frac{\sqrt{C*}}{8TL\sqrt{N}}$ and $\eta_g\geq 128L$, it follows that L must be a very small number, i.e., $L\leq\frac{1}{32\sqrt{T}}$, and as T increases, L must become increasingly smaller. This is a very strong assumption.
ii).The values of $\eta_g$ in Corollary 1 do not satisfy the condition $\eta_g \leq \frac{\sqrt{C*}}{8TL\sqrt{N}}$ in Theorem 1.

Besides, the numerical results for the full model version of SubDisMO should be added. Theoretically, the performance of this version should align with that of FedAvg and FedSAM, but these results are not included in Table 1. Adding this comparison would provide a more comprehensive evaluation.

**Questions:**

1.In Theorem 1, it appears that a smaller $\delta$ leads to a better bound, which is counterintuitive and contradicts the experimental performance. Could you further clarify the impact of $delta$?

2.Why do the performances of FedAvg and FedSAM in your experiments appear to be lower than those reported in the original paper? The original results show that both methods achieve over 80% accuracy on CIFAR-10, whereas your reported results are below 60%.

---

> ### Author Response · Authors · 2024-11-21
> **Response to Weakness and Question1**
>
> We thank the reviewer haCk for the time and valuable feedback! We would try our best to address the comments one by one.
>
> **Response to Weakness1:**
>
> Thank you very much for the insightful comments. We carefully revised the proof of Theorem 1 and noticed that in Appendix C the proof of Theorem 1 on page 14, the scaling of the inequality is indeed inappropriate, which leads to your first concern. And we attentively adjust the inequality and derive the learning rate conditions. Detailed process can be found on pages 14-19 of the revised manuscript. Thus, we can conclude that when the learning rates satisfy the following conditions, and **the assumption of constant $L$ can be cancelled**.
>
> $
> \begin{aligned}
> 16\eta_l^2L^2T^2\leq1\Rightarrow\eta_l\leq\frac{1}{4LT}\\\\
> 32\eta_l^2T^2\frac{N}{\mathcal{C}^*}L^2\leq \frac{1}{16} \Rightarrow \eta_l\leq\frac{\sqrt{\mathcal{C}^*}}{16TL\sqrt{N}}\\\\ 96L^3\eta_l^3\eta_gT^3\frac{N}{\mathcal{C}^*}\leq \frac{1}{16} \Rightarrow \eta_g\leq\frac{2\sqrt{N}}{\sqrt{\mathcal{C}^*}}\\\ 3L\eta_l\eta_g T\leq \frac{1}{16} \Rightarrow \eta_l\eta_g\leq\frac{1}{48TL}
> \end{aligned}
> $
>
> About your second concern, based on the above modified conditions of learning rates, we carefully choose $\eta_l = \frac{1}{\sqrt{Q}}$ and $\eta_g = \frac{\sqrt{\mathcal{C}^*}}{\sqrt{T}}$, and perturbation radius $\delta$ proportional to the learning rate, e.g., $\delta = \frac{1}{\sqrt{Q}}$, the convergence rate can be expressed as follows $\frac{1}{Q}\sum_{q=1}^Q \sum_{i \in \mathcal{K} _ q} \mathbb{E}\left[\|\nabla f^i(\theta _ q)\|^2\right] \leq \mathcal{O}\left(\frac{A _ 0}{\sqrt{QT\mathcal{C}^*}} + \frac{l^2 B _ 0}{\mathcal{C}^*} + \frac{\sigma _ g^2}{\mathcal{C}^*} + \frac{\sigma _ l^2}{TQ} + \frac{1}{\sqrt{TQ\mathcal{C}^*}}\right)$
> where $A _ 0 = \mathbb{E}[f(\theta_1)]$, $B _ 0 = \frac{1}{Q}\sum_{q=1}^Q \mathbb{E}[f(\theta _ q)]$. The main term of the revised bound is identical to the original Corollary 1.
>
> So, we can find that **the convergence rate is identical to that of the original manuscript,** $\mathcal{O}(\frac{1}{\sqrt{QT\mathcal{C}^*}})$. We have highlighted all the changes in the revised version, and we sincerely hope that you can find our response satisfactory.
>
>
> **Response to Weakness2:**
>
> Thank you very much for the insightful comments. As we mentioned on Page 3, the proposed SubDisMO gives a new insight into solving distributed minimax optimization, especially under resource-limited scenarios. When $\mathcal{C}^*=N$ that is all the clients train the full model, SubDisMO achieves the same convergence rate $\mathcal{O}(1/\sqrt{QTN})$ as FedSAM. That is **FedSAM is a special case of our SubDisMO** when each client trains the full model. And the results of FedSAM are reported in the Page 8, Table 1 from the original manuscript.
>
> **Response to Question1:**
>
> Thank you very much for your comments.
> (1) Indeed, as stated in Theorem 1, a smaller value of $\delta$ leads to a tighter bound. However, it is important to note that this bound is about the averaged gradient. **A smaller averaged gradient does not necessarily imply better experimental performance.** Instead, it indicates that the algorithm can converge more quickly to a critical point. (2) Furthermore, in Section 5.3, we explore the impact of $\delta$, and give the empirical results. It can be seen that the performance would not definitely improve as $\delta$ gets smaller, varying with different datasets. Thus, it is necessary to select the appropriate $\delta$ through empirical experiments.

---

> > ### Comment · Reviewer_haCk · 2024-11-27
> >
> > My concern on the parameter $\eta_l$, $\eta_g$ and $L$ are partially addressed. However, I still have the following concerns.
> >
> > 1. You have fixed the proof, and I tried to follow it. However, as you have local update on each node, outer loop iteration and inner loop iteration. It really hard to tell what you referred exactly in the proof, say $\theta_q$, $\tilde \theta_q$ and $\Delta_q^i$ in line. 736-747. I have check both the main text and the appendix, NO exact definition of these variables but some similar definitions $\theta_q^i$ and $\Delta_{q,n}^i$.  Maybe it is obvious, however, as a theoretical paper, the authors must pay more attention to these details and make. these exact, instead of letting the readers to guess.
> >
> > 2. In your analysis, $delta$ would only introduce extra errors. However, the empirical results indicate that the perturbation. is important in achieving the SOTA result. You show $\delta = 0.01$ Acc= 49.73 v.s. $\delta$ = 0.1 Acc=51.23 in Table 2, while in column 2 of Table 1 many baseline method achieve Acc higher than 49.73. There is a great gap here. The empirical performance improvement comes from extra errors in theory. I am confused.
> >
> > Considering all these, I suggest the authors check the theory and experiments of their proposed method and remain my score.

---

> > > ### Author Response · Authors · 2024-11-27
> > >
> > > Thank you very much for your comments.
> > >
> > > **Response to Question 1:**
> > >
> > > We're sorry for the unclear definition.
> > >
> > > $\theta_q$ is the global full model in communication round $q$.
> > >
> > > $\tilde{\theta}_q$ is the perturbed global full model in communication round $q$.
> > >
> > > $\Delta_q^i$ is the accumulated updates for parameter $i$ of global full model in communication round $q$.
> > >
> > > All the definitions of the above notations have been updated in the notations table in Appendix A in the revised version.
> > >
> > > **Response to Question 2:**
> > >
> > > The introduction of the parameter $\delta$ is important, which can enhance the generalization of our method. $\delta$ restricts the scope of the perturbation which further affects the direction of gradient descent.
> > >  We utilize $\delta$ and $\tilde{\theta} _ {q,n,t-1}=\theta _ {q,n,t-1}+\delta\frac{g _ {q,n,t-1}}{\parallel g _ {q,n,t-1}\parallel}$ to gain the perturbed local submodel.
> > > Thus **it's necessary to select the appropriate $\delta$ through empirical experiments to gain better performance**. **In FedSAM[Qu et al.,2022], authors also discuss the impact of $\delta$ on full model**. It can be seen that inappropriate $\delta$ leads to performance degradation even also worse than other baselines.
> > >
> > > In addition,  we consider that $\delta$ introduces additional errors in the **averaged gradient**, which primarily impacts the **convergence speed** rather than the accuracy performance.
> > >
> > > We will be very happy to clarify further concerns (if any). If we have resolved all your concerns, we would greatly appreciate it if you could raise the score.

---

> ### Author Response · Authors · 2024-11-21
> **Response to Question2**
>
> Thank you very much for the insightful comments.
>
> (1) First, the original paper of FedAvg and FedSAM that you mentioned all utilized the convolution architecture model, such as CNN and ResNet, whose model architecture is quite different from ours (ViT, Transformer models). In our work, we employed ViT-small and ViT models on CIFAR-10 and CIFAR-100. We newly conduct experiments using ResNet18 on CIFAR-10 with Non-IID data distribution Dir($\mu=1.0$). The new results are shown as follows and our proposed method SubDisMO achieves 76.34\% accuracy and outperforms all the baselines with submodels.
>
> (2) Additionally, we have trained using both transformer-based architectures and convolution-like model ResNet in a centralized way, which can be considered as the upper bound of the experimental performance. We can see that our proposed method SubDisMO can achieve a satisfactory level compared with the centralized models.
>
> |**Methods**|**ResNet18/CIFAR-10**|**ViT-Small/CIFAR-10**|**ViT/CIFAR-100**|
> |:--:|:--:|:--:|:--:|
> |Centralized|81.02|59.19|35.61|
> |FedAvg|76.78(2.92)|55.70(2.96)| 30.95(2.90)|
> |FedSAM|78.35(2.89)|57.03(2.62)| 32.11(2.68)|
> |IST|62.51(5.39)|38.06(10.27)|17.15(5.54)|
> |OAP|64.90(3.54)|48.29(9.29)|26.09(6.97)|
> |PruneFL|64.04(4.06)|48.20(5.16)|22.35(5.80)|
> |FedRolex| 65.30(3.16)|44.84(4.75)|21.73(2.57)|
> |RAM-Fed|69.15(3.15)|50.19(4.16)|23.25(4.77)|
> |SubDisMO|**76.34(3.33)**|**51.23(4.77)**|**25.43(4.56)**|
>
> If there are any further questions, we are happy to clarify and try to address them. Thank you again and your recognition means a lot for our work.

---

### Meta-Review · Area_Chair_AAXJ · 2024-12-21

**Metareview:**

The paper presents SubDisMO, a resource-aware distributed minimax optimization algorithm, designed to address challenges in resource-limited clients while maintaining generalization. The work offers a novel perspective by proposing submodel pruning and perturbation techniques to enhance generalization and achieve optimal convergence rates. The theoretical analysis is sound, and experiments on CIFAR-10 and CIFAR-100 provide evidence of the method’s advantages over several baselines. Strengths of the paper include the originality of combining distributed optimization with resource awareness, the detailed theoretical results, and a practical approach to addressing resource limitations in large-scale systems.

However, despite these merits, there are significant concerns. During the discussion phase, some reviewers attempted to evaluate the reproducibility of the results using the anonymized code repository provided by the authors. Unfortunately, the repository returned an error stating, “The repository is expired,” which prevented verification of key claims. Additionally, while the theoretical contributions are promising, the explanations of the theorems and assumptions in the main paper could be significantly improved for readability and accessibility. Important parameter definitions were unclear, and certain assumptions, such as those in Theorem 1, remain overly restrictive. Furthermore, empirical results exhibit inconsistencies, particularly regarding the random and rolling masking policies, which raise further questions about the robustness of the proposed method. We encourage the authors to address these issues thoroughly and resubmit in the future.

**Additional Comments On Reviewer Discussion:**

During the discussion phase, several important points were raised about the submission. Concerns included unclear parameter definitions and inconsistencies in theoretical assumptions, such as those in Theorem 1. Questions were also raised about the formulation of the distributed minimax optimization problem and the need for additional analysis on masking policies. There were further requests for clarification on scalability and computational complexity comparisons. Some reviewers expressed interest in gaining deeper insights from the provided code repository to address specific doubts; however, the repository returned an error stating it had expired, preventing further validation of the claims. While the authors provided detailed responses, including revised theoretical proofs, added parameter definitions, and additional experiments with rolling masking, key issues remained unresolved. These included inconsistencies in empirical results, unclear explanations in the main paper, and the inability to verify results through the expired repository.

---

### Decision · Program_Chairs · 2025-01-22

Reject